# A lentiviral vector for the production of T cells with an inducible transgene and a constitutively expressed tumour-targeting receptor

Patrick Reichenbach[1,2], Greta Maria Paola Giordano Attianese[1,2], Khaoula Ouchen[1], Elisabetta Cribioli[1], Melanie Triboulet[1], Sarah Ash[1], Margaux Saillard [1], Romain Vuillefroy de Silly [1], George Coukos [1] ✉ & Melita Irving [1] ✉

Vectors that facilitate the engineering of T cells that can better harness endogenous immunity and overcome suppressive barriers in the tumour microenvironment would help improve the safety and efficacy of T-cell therapies for more patients. Here we report the design, production and applicability, in T-cell engineering, of a lentiviral vector leveraging an antisense configuration and comprising a promoter driving the constitutive expression of a tumour-directed receptor and a second promoter enabling the efficient activation-inducible expression of a genetic payload. The vector allows for the delivery of a variety of genes to human T cells, as we show for interleukin-2 and a microRNA-based short hairpin RNA for the knockdown of the gene coding for haematopoietic progenitor kinase 1, a negative regulator of T-cell-receptor signalling. We also show that a gene encoded under an activation-inducible promoter is specifically expressed by tumour-redirected T cells on encountering a target antigen in the tumour microenvironment. The single two-gene-encoding vector can be produced at high titres under an optimized protocol adaptable to good manufacturing practices.

Important technological advances in recent years in the field of cellular engineering have enabled increasing clinical translation of gene-modified cells for the treatment of cancer and other diseases[1–6]. Transient or stable alterations can be made to host cells, such as haematopoietic stem cells[7], or immune cells including T cells[8], B cells[9], natural killer cells[10] and macrophages[11], to modify their properties for a desired therapeutic outcome upon re-infusion into a patient. Disruption of cellular processes can be attained by silencing, correcting or overexpressing targets within the genome, or by RNA interference of transcribed genes such as by short hairpin (sh)RNA or microRNA (miR; non-coding RNAs)[12]. If only temporary changes in gene expression are desired, such as for evaluating the safety of a previously untested cellular product, messenger (m)RNA electroporation can be used[13], and advances in non-viral episomal vector design show promise in enabling longer-term modifications to gene expression[14,15]. For permanent modifications, a variety of tools have been developed for genome editing

[1]Department of Oncology, Ludwig Institute for Cancer Research Lausanne, Lausanne University Hospital and University of Lausanne, Lausanne, Switzerland. [2]These authors contributed equally: Patrick Reichenbach, Greta Maria Paola Giordano Attianese. ✉e-mail: George.Coukos@chuv.ch; Melita.Irving@unil.ch

including zinc finger nucleases[16], transcription activator-like (TAL) effector nucleases[17], clustered regularly interspaced short palindromic repeats (CRISPR)/Cas9[18,19] and viral vectors such as adenovirus, adeno-associated virus (AAV)[20] and retroviruses[21–23].

Both lentivirus and gamma-retrovirus are subtypes of retrovirus comprising an RNA genome that is converted to DNA in infected host cells by the virally encoded enzyme reverse transcriptase[7], and they allow efficient non-site-directed integration of genes of interest into the genome[21]. Lentiviral and gamma-retroviral vector-based gene-engineering strategies have been widely and safely used in the clinic for both chimeric antigen receptor (CAR)- and T cell receptor (TCR)-T-cell therapy of cancer[23]. In particular, CAR-T cells targeting the B-cell lineage antigen CD19 have conferred unprecedented clinical responses against certain haematological malignancies, such as acute lymphoblastic leukaemia. In addition, TCR-engineered T cells targeting the HLA-A2-restricted cancer testis epitope NY-ESO-1$_{157-165}$ (A2/NY) have shown promise for the treatment of melanoma, myeloma and synovial cell sarcoma[24–27]. The continued importance of lentiviral vectors as a tool for T-cell engineering purposes for clinical application is underscored by recent advances in improving CAR-T-cell manufacturing protocols[28].

CARs are synthetic receptors that can be used in place of a TCR-CD3 complex to link tumour-antigen binding and cellular activation upon target engagement in a non-major histocompatibility complex (MHC)-restricted manner. While first generation (1G) CARs comprise the endodomain of CD3-zeta for signal 1 of T-cell activation, 2G and 3G CARs further include one or more co-stimulatory endodomains, respectively. As previously mentioned, CAR therapy has been a powerful strategy for fighting some advanced haematological malignancies, but a considerable proportion of patients either do not benefit or experience relapse. Moreover, epithelial-derived solid tumours remain poorly responsive[8] to CAR therapy, and the efficacy of TCR-engineered T cells[25], as well as of tumour-infiltrating lymphocyte transfer, have proven beneficial against relatively few cancer types in a modest proportion of patients[29]. It is widely held, however, that the development of personalized combinatorial or/and co-engineering strategies to overcome barriers in tumour microenvironment (TME) and harness endogenous immunity can further improve responses to these different T-cell-based therapies[30–32]. Co-engineered CAR-T cells are referred to as 4G CARs, armoured CARs or next-generation CARs, and the term TRUCK ('T cells redirected for universal cytokine mediated killing')[33] has been coined to define T cells specifically engineered to enforce expression of cytokines/interleukins (ILs). Examples of cytokines evaluated in the context of CAR- and TCR-T cells, and in some instances tumour-infiltrating lymphocytes, include IL-12[32,34], IL-15[23,35] and IL-18[36,37].

While in early studies the co-expression of genes in T cells was achieved by dual transduction[38,39], the high cost of good manufacturing practice (GMP)-grade virus production and elevated risk for insertional mutagenesis[40] have driven the development of 'all-in-one' multi-gene encoding vectors[41,42]. If both the receptor (CAR or TCR) and the gene cargo are constitutively expressed, they can be separated on the transfer vector by an internal ribosome entry site (IRES)[43]. Alternatively, for equimolar expression of both genes, a picornavirus 2A peptide sequence (P2A)[44,45] can be used. For both approaches, RNA is generated from a single promoter and co-expression is reliant upon functioning of the interspersed element. Disadvantages of IRES are its relatively large size (about 500 bp), cell-type dependency[46] and reduced expression of the downstream gene[43]. Drawbacks of P2A are the risk of incomplete cleavage and potential immunogenicity of the gene product[47].

To minimize the risk of systemic toxicity and enhance T-cell function, it may be preferable to limit expression of the gene cargo to the TME. One approach to achieve this is to place the gene cargo under a T-cell-activation-dependent promoter such as nuclear factor of activated T cells (NFAT) response elements fused to the IL-2 minimal promoter (6xNFAT)[48,49]. Here we demonstrate that previously described dual promoter sense and bidirectional vectors are limited by interference of gene expression[12] and promoter leakiness, respectively, in transduced cells. We subsequently present a dual inverted promoter vector design, along with an optimized protocol for the production of high-titre lentiviral particles to overcome the aforementioned obstacles. Overall, our antisense gene-cassette design and methodology for lentivirus vector production have important implications for improving the performance and safety of engineered T cells for cancer immunotherapy. Moreover, our approach can be considered universal as it can be applied to other vector types and different gene therapies.

## Results

### Antisense vector design to accommodate independent promoters

Here we sought to optimize lentivirus vector-mediated independent co-expression of two genes in transduced human T cells, with one gene under a constitutive promoter and the other under an inducible promoter, to improve adoptive T-cell transfer (ACT) of cancer. We began by building a panel of transfer vectors comprising the promoters in dual sense and bidirectional orientations (Fig. 1a,b, left). For our study, we selected the constitutive human phosphoglycerate kinase (PGK) promoter for gene A, and 6xNFAT for gene B. For screening purposes, we placed *egfp* under PGK and *mCherry* under 6xNFAT (lentivirus vector component sequences are found in Supplementary Table 1).

The production of second-generation lentivirus vectors relies on the co-transfection of: (1) a transfer, (2) a packaging and (3) an envelope vector into a producer cell line such as human embryonic kidney (HEK)293T cells (that is, HEK293 cells expressing the oncogenic SV40 large T-antigen thought to promote plasmid-mediated gene expression)[50]. Lentiviral vectors typically comprise three HIV-1 genes: (1) *gag* (which is processed to matrix and other retroviral core proteins) and (2) *pol* (reverse transcriptase, RNase H and integrase functions), both found on the packaging plasmid, as well as (3) *env* (envelope protein that resides in the lipid bilayer and determines viral tropism) on the envelope vector. We have used the vesicular stomatitis virus G-protein (VSV-G) pseudotype[51], which broadens the type of cells that can be infected[52] as compared with the HIV envelope[53]. Notably, the transfer vector does not encode viral sequences, except for necessary *cis*-acting sequences such as the long terminal repeat (LTR), packaging signals and the Woodchuck hepatitis virus post-transcriptional regulatory element (WPRE) to enhance expression of the transgene[54]. The LTRs, located at each end of the provirus, comprise U3, R and U5 regions and function as a eukaryotic transcription unit. Specifically, the U3 region contains the viral promoter and enhancer elements, the R region includes the mRNA initiation site, and the U5 region is involved with polyadenylation. Notably, the 3'LTR of the transfer vector has been truncated (U3 has been removed) to generate self-inactivating lentivirus vectors[55].

Here, to produce lentiviral particles, HEK293T cells were transfected with lentiviral packaging and envelope plasmids, along with differently designed transfer vectors, and crude supernatant was used directly to transduce Jurkat cells. For the sense transfer vector configuration, the 6xNFAT promoter and gene B (*mCherry*) were placed in the same orientation upstream of the PGK promoter and gene A (*egfp*) (Fig. 1a, top left). Indeed, the inducible promoter cannot be placed downstream of the constitutive one as there will be readthrough, and hence constitutive expression, of both genes by the upstream promoter. Moreover, it is not possible to place a polyadenylation (PA) site between the two genes to avoid interference because this will abrogate virus production in the HEK293T cells (depicted in Fig. 1a, bottom left).

We evaluated expression of dual sense orientation genes as described above in unstimulated and stimulated Jurkat cells. We observed expression of EGFP in unstimulated Jurkat cells, and co-expression of both EGFP and mCherry upon stimulation (Fig. 1a, right). For the latter, transcription of both genes must reach the same

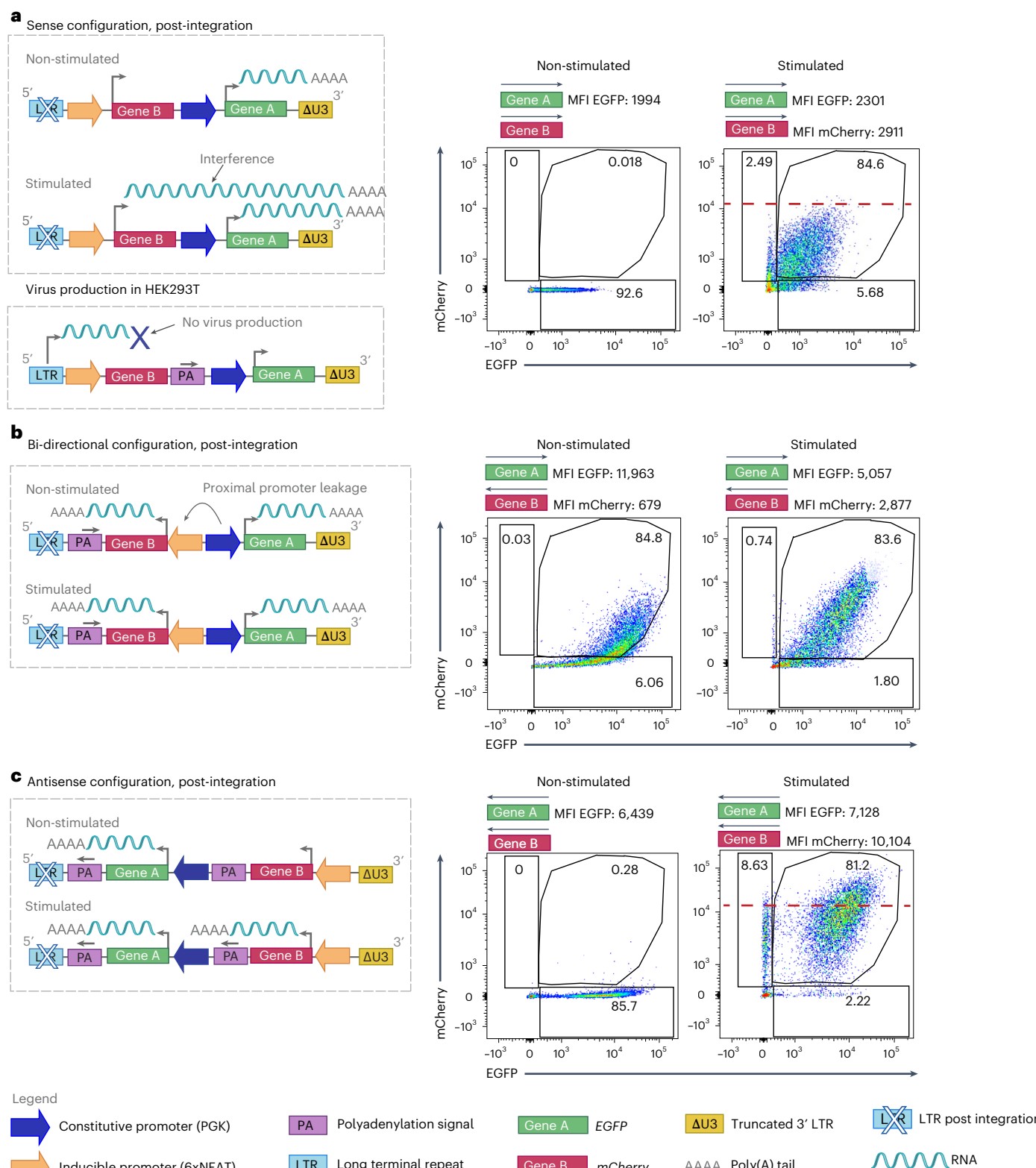

**Fig. 1 | Dual antisense lentiviral transfer vector allows efficient constitutive expression of a transgene and controlled co-expression of an activation-inducible transgene.** For all dual transfer constructs, EGFP (Gene A) expression is constitutively driven by the PGK promoter and mCherry (Gene B) by 6xNFAT. **a**, Left: schematic of dual sense orientation lentiviral transfer vector post-integration in non-stimulated (top) and stimulated (middle) transduced cells. Left, bottom: schematic illustrating that the inclusion of a PA site between the 2 genes will abrogate virus production in the packaging cells. Right: representative flow cytometric analysis of transfected Jurkat cells pre- and post-stimulation. **b**, Left: schematic of bidirectional transfer vector post-integration in non-stimulated (top) and stimulated (bottom) transduced cells. Right: representative flow cytometric analysis of transduced Jurkat cells, pre- and post-stimulation. **c**, Left: schematic of antisense orientation lentiviral transfer vector post-integration in non-stimulated (top) and stimulated (bottom) transduced cells. Right: representative flow cytometric analysis of transduced Jurkat cells, pre- and post-stimulation. The dashed red line demarcates the increase in mCherry-EGFP MFI for dual antisense versus sense configuration vectors in stimulated Jurkat cells. The flow cytometry plots are representative of 5 independent experiments.

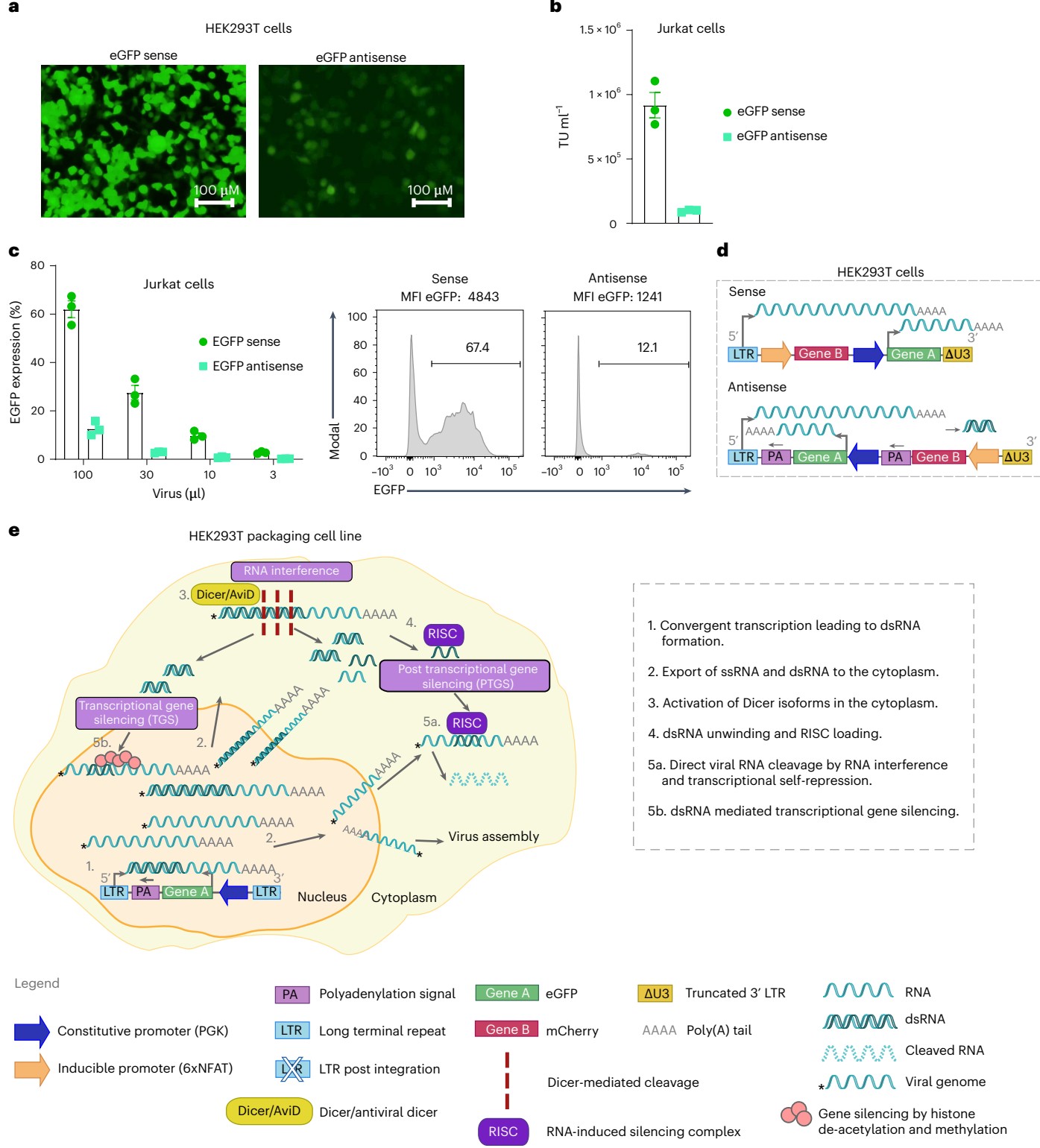

**Fig. 2 | Antisense transfer vector yields lower lentivirus vector titre than sense vector.** For all dual constructs, EGFP (Gene A) expression is constitutively driven by the PGK promoter and mCherry (Gene B) by 6xNFAT. **a**, Representative microscopy images (×10 magnification) of HEK293T cells transfected with dual sense (left) vs antisense lentiviral vectors (right) for lentivirus vector production. **b**, Viral titres (TU ml⁻¹). **c**, Transduction of Jurkat cells with decreasing volumes of lentivirus vector supernatant to evaluate %EGFP expression (on day 5) by flow cytometry. Bar graphs represent the mean ± s.e.m. of technical duplicates for 3 independent experiments. Representative histograms of transduction with 100 μl virus supernatants are shown for dual sense (left) and antisense (right) approaches. **d**, Schematic of dual sense (top) vs antisense (bottom) orientation lentiviral transfer vectors encoding both EGFP and mCherry. **e**, Illustration of potential Dicer-associated mechanisms in response to dsRNA, which may be limiting to lentivirus vector production in HEK293T cells.

3' LTR for polyadenylation to occur and it has been previously reported that this configuration can cause transcriptional interference which limits transgene expression[56,57]. Indeed, interference resulting from the dual sense configuration is evident upon comparison of the mean fluorescence intensity (MFI) for mCherry when encoded alone versus upstream of constitutively expressed EGFP (Extended Data Fig. 1a).

To avoid such interference, we next evaluated a bidirectional configuration (Fig. 1b, left) in which the orientation of Gene B and its promoter are inverted. Notably, for inverted Gene B, no longer restricted by polyadenylation at the LTR, we employed an inverted bovine growth hormone (BGH) PA site[58]. Of note, an inverted PA site will not interfere with virus production. However, despite the separation of the two-gene cassettes, we observed leakage from the inducible promoter as evidenced by mCherry expression in non-activated Jurkat cells, presumably due to the proximity of strong enhancer elements of the constitutive promoter (Fig. 1b, right).

Finally, to prevent both interference and leakage issues as seen for the first two transfer vector designs, we built a dual antisense configuration vector (Fig. 1c, left) in which Gene A has its own PA signal derived from BGH, and Gene B is followed by a synthetic polyadenylation site (SPA) and a human transcription pausing site (to prevent transcriptional readthrough)[57]. We observed the highest level of expression of both EGFP and mCherry in activated Jurkat cells among the 3 configurations evaluated, and there was no mCherry expressed in non-activated Jurkat cells. For example, in the representative experiment shown in Fig. 1, in stimulated Jurkat cells, an MFI for mCherry of 10,104 was observed for the antisense configuration (Fig. 1c, right) vs an MFI of 2,911 for the sense configuration vector (Fig. 1a, right). While absolute MFI values varied between independent assays, within a given experiment we consistently observed a higher MFI for both EGFP and mCherry in activated Jurkat cells transduced with the dual antisense in comparison with the dual sense lentiviral vector (Extended Data Figs. 1 and 2). This is probably due to the lack of transcriptional interference as well as the use of the BGH PA site, which is stronger than polyadenylation by the LTR[58]. We thus continued our study with this dual inverted transfer vector configuration.

### Overcoming low lentiviral titres by abrogating the anti-dsRNA response

Post-integration, the dual antisense vector configuration enabled the best co-expression of both a constitutive and an inducible gene in transduced activated Jurkat cells (that is, no competition to reach the PA site, no leakiness by the inducible promoter and highest MFI of both EGFP and mCherry post-activation) (Fig. 1 and Extended Data Figs. 1 and 2). However, during lentivirus vector production, we observed an obvious decrease in EGFP expression levels for vectors comprising the dual antisense vs sense orientation of the transgenes (Fig. 2a), which corresponded to much lower viral titres for the antisense lentiviral vectors (Fig. 2b). Indeed, transduction of Jurkat cells with 100 μl lentiviral supernatant yielded about 60% transduction efficiency for the dual sense orientation vector vs about 10% (and lower MFI) for the dual inverted vector (Fig. 2c). Similarly, for single gene cassettes, lower viral titres were observed for antisense vs sense lentiviral vectors (Extended Data Fig. 3a).

Hence, we next sought to overcome barriers to the production of lentiviral particles comprising an antisense transfer vector. During lentivirus vector production in HEK293T cells, both the 5'LTR and the inverted PKG promoter of the antisense vector are active, thus resulting in the generation of double-stranded (ds)RNA by convergent transcription (as illustrated in Fig. 2d). Although intracellular innate immunity may be triggered in response to dsRNA upon detection by nuclear and cytosolic sensors such as during a natural viral infection, this has been shown not to limit lentivirus vector titre because HEK293T cells do not generate an interferon (IFN) response. Indeed, it has recently been revealed that HEK293T as well as various stem-cell-like lines employ an RNA interference (RNAi) response involving various Dicer isoforms upon detection of dsRNA[59,60]. We thus postulated that the dsRNA resulting from convergent transcription[61] may be subject to Dicer and/or Dicer isoform-mediated (for example, aviD) cleavage within the nucleus or cytoplasm and that small interfering (si)RNA products created during this process are involved either in RNAi-mediated self-degradation or/and in transcriptional gene silencing of the viral RNA to be packaged[62] (as illustrated in Fig. 2e).

We devised two approaches to overcome these potential barriers to lentivirus vector production arising from convergent transcription, the first being to inhibit the antiviral RNAi machinery to prevent disruption of the viral genome by taking advantage of a natural viral mechanism to evade immunity. Specifically, Nodamuravirus expresses an RNA interference suppressor protein called B2 (hereafter referred to as NovB2)[60,63] and it has been previously utilized to increase viral titres of bidirectional vectors by at least fivefold[61] via inhibition of Dicer isoforms[60,64]. We hence took the strategy of co-expressing NovB2 from the envelope vector (Fig. 3a) and achieved an important increase in viral titre (Fig. 3b). Indeed, we observed a fivefold rise in the proportion of EGFP+ Jurkat cells upon transduction with dual antisense lentivirus vector (Fig. 3c). The use of NovB2 also increased titres for single gene cassette inverted lentiviral vectors (Extended Data Fig. 3b).

### Overcoming low lentiviral titres favouring transcription of the viral genome

For our second approach to improve lentivirus vector titres, we sought to favour the transcription of the viral genome for packaging (that is, single stranded (ss)RNA transcription from the 5' LTR) by exploiting the human T-cell leukaemia virus 1 Tax protein. The Tax protein[65] is associated with the transcriptional promotion of viral proteins (including in the nucleus during infection), and the regulation of many signalling pathways including CREB/ATF, NF-κB, AP-1 and RSF[66]. To test whether Tax could be used to increase viral titres[65], we replaced the initial Rous sarcoma virus (RSV)-based promoter and enhancer region at the 5' LTR with the cytomegalovirus (CMV) promoter and enhancer which comprises 4 consensus NF-κB binding motifs[67] (schematic in Fig. 3d). We then produced virus in the presence or absence of co-transfected Tax-expressing plasmid (Fig. 3d). We observed a similar gain in titre, transduction efficiency and transgene expression levels (MFI) as achieved in the context of NovB2 (Fig. 3e–g). It is likely that the Tax-mediated increase in lentivirus vector titre is due to a change in stoichiometry in favour of viral genome transcript, as well

**Fig. 3 | Rescue of low dual antisense vector lentiviral titres in the presence of NovB2 and Tax proteins.** For dual constructs, EGFP (Gene A) expression is constitutively driven by the PGK promoter and mCherry (Gene B) by 6xNFAT. **a**, Schematic of dual sense vs antisense orientation lentiviral transfer vectors encoding both EGFP and mCherry. Antisense transfer lentivirus vector was produced in the presence or absence of NovB2 (encoded on the envelope plasmid). **b**, Viral titres (TU ml⁻¹). **c**, Left: transduction of Jurkat cells with decreasing volumes of lentivirus vector supernatant to evaluate % EGFP expression (on day 5) by flow cytometry. Bar graphs represent the mean ± s.e.m. of 3 independent experiments. Right: representative histograms for Jurkat cells transduced with 100 μl of lentivirus vector supernatant produced in the absence or presence of NovB2. **d**, Left: schematic of dual antisense vector encoding EGFP and comprising a chimeric LTR (ΔU3, R and U5) for which the RSV promoter and enhancer at the 5' LTR has been substituted by the complete CMV promoter and enhancer. Right: schematics representing antisense lentivirus vector production in the presence or absence of Tax protein (via vector co-transfection), or of NovB2 (encoded on the envelope plasmid), or of both Tax and NovB2. **e**, Transduction of Jurkat cells with decreasing volumes of lentivirus vector supernatant to evaluate % EGFP expression (on day 5) by flow cytometric analysis. Bar graph shows the mean ± s.e.m. of 3 independent experiments. **f**, Viral titres (TU ml⁻¹). **g**, Representative histograms of Jurkat cells transduced with 30 μl of lentiviral supernatant.

as higher transcription of the packaging and envelope vectors which also comprise CMV promoters. Finally, we observed that Tax and NovB2 were able to act jointly to restore antisense viral titres, transduction efficiency and levels of transgene expression (MFI) (Fig. 3e–g).

### Inducible gene cargo encoded in antisense is efficiently expressed upon T-cell activation in vitro

We next sought to test our dual inverted vector design and optimized methodology for lentivirus vector production in the context of both

next-generation (4G) CAR- and TCR-T cells. For proof-of-principle, we began by constructing vectors comprising an anti-PSMA or anti-CD19 CAR (constitutively expressed under PGK)[68], along with luciferase as inducible gene cargo (under 6xNFAT) (Fig. 4a). We also generated an equivalent sense orientation transfer vector for the anti-PSMA CAR and luciferase. Lentivirus vector was produced in the presence of NovB2 and Tax and we observed that both human CD4$^+$ and CD8$^+$ T cells were efficiently transduced with the 4G constructs (Fig. 4b). To achieve an equivalent percentage of 4G CAR$^+$ T cells for functional testing,

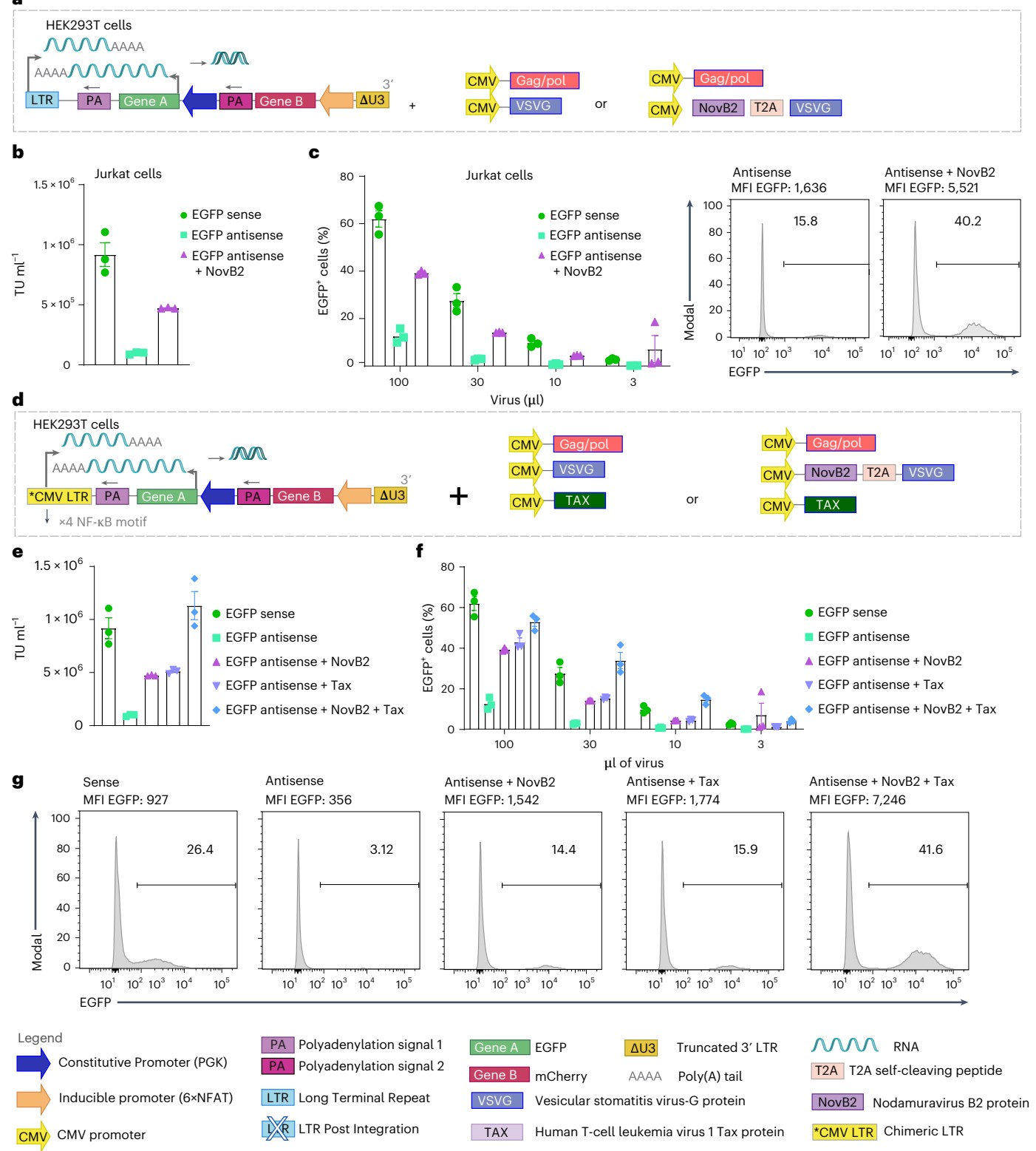

the transduced T cells were mixed with untransduced (UTD) T cells to reach 40% CAR+ (that is, the lowest transduction efficiency as achieved for CD8+ T cells with the 4G anti-CD19 CAR, Fig. 4b). The 4G CAR-T cells all efficiently and specifically killed target cells in co-culture assays (Fig. 4c, left and right). While there were no differences in specific target-cell killing by the 4G anti-PSMA CAR-T cells generated with sense vs antisense lentiviral vectors, significantly higher levels of luciferase mediated luminescence were observed for the antisense design (Fig. 4d).

We further compared sense and antisense lentiviral transfer vectors encoding the anti-PSMA CAR and mCherry as inducible gene cargo. Once again, we produced lentivirus vector in the presence of NovB2 and Tax and achieved efficient transduction of both human CD4+ and CD8+ T cells (Fig. 4e, left). We further observed a significantly higher MFI for CARs expressed from the dual antisense vs sense lentiviral vectors (Fig. 4e, right). In line with our findings above, we observed no differences in cytotoxicity of target PC3-PIP tumour cells by anti-PSMA CAR-T cells generated with the different orientation lentiviral vectors (Fig. 4f, left). It is possible that more stringent conditions, such as the use of a weaker CAR, the co-culture of fewer CAR-T cells to target cells or the use of tumour cells with lower levels of target antigen, may reveal lower relative activity levels of CAR-T cells generated with the sense lentiviral vector. Upon T-cell activation in co-culture assays, mCherry expression levels steadily increased over time for the antisense lentiviral vector-generated 4G CAR-T cells, but mCherry was not detectable for the sense lentiviral vector-engineered CAR-T cells, even at 16 h (Fig. 4f, right). To confirm that this lack of detection was a sensitivity issue for the IncuCyte instrument-based assay rather than a defect in the sense vector, we evaluated the 4G CAR-T cells following 24 h of co-culture without and with target cells by flow cytometric analysis. We observed higher background levels of mCherry expression for the antisense vector (both percentage and MFI) in non-activated 4G CAR-T cells (Fig. 4g). However, we achieved similar transduction efficiencies for both the sense and antisense vectors as evidenced by the percentage of T cells expressing mCherry upon T-cell activation (that is, the sense orientation lentiviral vector is functional; Fig. 4h, left), and the MFI for mCherry upon activation was significantly higher for the antisense lentiviral vector (Fig. 4h, right). Significantly higher levels of mCherry expression (MFI) were also observed upon phorbol myristate acetate (PMA)-Ionomycin stimulation of the antisense lentiviral vector-generated 4G CAR-T cells (Fig. 4i).

Subsequently, we developed lentiviral transfer vectors encoding a clinically relevant HLA-A2-restricted NY-ESO-1$_{157-165}$ specific TCR[69] along with either IL-2 or mCherry as inducible gene cargo (Extended Data Fig. 4a). Lentivirus vectors encoding the TCR and IL-2 were produced in the presence of NovB2 and Tax, and human CD4+ and CD8+ T cells were efficiently transduced (Extended Data Fig. 4b). As for the CAR-T cells, equivalent percentages of TCR+-T cells were generated by appropriate mixing with UTD-T cells for all comparative functional assays. We observed similar levels of target-cell killing (Extended Data Fig. 4c) as well as IFNγ production (Extended Data Fig. 4d) upon co-culture with A2+/NY+ Saos-2 target cells for the IL-2 co-engineered TCR-T cells generated either with sense or antisense vectors. However, significantly higher levels of IL-2 were produced by the next-generation TCR-T cells generated with the antisense vs sense vector upon co-culture with target cells (Extended Data Fig. 4e). Differences in IL-2 gene cargo expression levels were not observed upon PMA-Ionomycin stimulation of the engineered T cells, a condition that drives the maximum production of endogenous IL-2 (Extended Data Fig. 4f). Finally, TCR-T cells with inducible mCherry as gene cargo generated from antisense vs sense vectors were tested. Upon T-cell activation in co-culture assays with target cells, an increase in mCherry was evident over time for the antisense but not for the sense lentiviral vector-generated TCR-T cells (Extended Data Fig. 4g). However, flow cytometric analysis of next-generation TCR-T cells following 24 h co-culture with A2+/NY+ Saos-2 target tumour cells confirmed that mCherry was in fact produced by T cells generated with both antisense and sense lentiviral vectors (Extended Data Fig. 4h, left) but that mCherry expression levels (MFI) were very low for the sense orientation (Extended Data Fig. 4h, right). Significantly higher levels of mCherry expression (MFI) were also observed upon PMA-Ionomycin stimulation of the antisense vs sense lentiviral vector-generated TCR-T cells (Extended Data Fig. 4i).

## Inducible gene cargo encoded in antisense is efficiently expressed upon T-cell activation in vivo

For in vivo proof-of-principle of our antisense lentiviral vector approach, we evaluated next-generation anti-PSMA and anti-CD19 CAR-T cells with luciferase (for imaging purposes) expressed under 6xNFAT as inducible gene cargo (Extended Data Fig. 5a). Efficient transduction of primary human T cells was achieved for both antisense lentiviral 4G CAR constructs (Extended Data Fig. 5b, left). We observed low levels of background mCherry expression in non-activated anti-CD19 CAR-T cells, presumably due to minor tonic signalling[70], but upon

**Fig. 4 | In vitro testing reveals higher activation-induced expression levels of gene cargo by 4G CAR-T cells engineered with antisense vs sense lentiviral vectors. a**, Schematic of lentiviral vectors encoding an anti-PSMA or anti-CD19 2G CAR (gene A) under the PKG promoter and luciferase or mCherry as gene cargo (gene B) under 6xNFAT, in both sense and antisense configurations. The 2G CARs comprise a tumour-targeted scFv, the linker region of CD8α, the TM and ED of CD28, and the ED of CD3z. **b**, Transduction efficiency of primary human CD4+ and CD8+ T cells with the 2 different CARs and luciferase constructs as measured by cell-surface CAR staining on day 9. Shown are mean ± s.e.m. for T cells from 3 independent healthy donors. **c**, PSMA+ PC3-PIP (right) or PC3-CD19+ engineered tumour cell lines (left) killing assay by the CAR-T cells and UTD-T cells as measured by the IncuCyte instrument (decrease in total green area per μm² corresponds to target-cell death) over time. Shown are mean ± s.e.m. Symbols indicate individual donors (n = 3). (NS, not significant, P = 0.9173 sPSMA vs aPSMA; **P = 0.0049 aPSMA vs UTD, P = 0.0507 sPSMA vs UTD, **P = 0.0025 aCD19 vs UTD). Statistical significance was assessed using two-way ANOVA and post-hoc Tukey test vs UTD. **d**, Evaluation of luciferase expression levels (luminescence (counts)) by activated anti-PSMA- (left) and anti-CD19-CAR-T cells (right), measured by HIDEX. Values for assay are the mean ± s.e.m. for n = 3 human T-cell donors. (*P = 0.0484 aPSMA vs tumour control; ***P < 0.001 aCD19 vs tumour control). Statistical significance was assessed using one-way ANOVA vs tumour cells alone. **e**, Transduction efficiency of primary human CD4+ and CD8+ T cells. Left: percentage of CAR+ positive cells. Right: MFI of positive cells by direct surface cell staining on day 9 (*P = 0.0447 CD4+ sPSMA vs aPSMA; *P = 0.0229 CD8+ sPSMA vs aPSMA). Values for assay are the mean ± s.e.m. for n = 3 human T-cell donors. Statistical significance was assessed using one-way ANOVA. **f**, Left: PSMA+ PC3-PIP killing assay by CAR- and UTD-T cells as measured by the IncuCyte instrument (total green area per μm²) over time. Shown are mean ± s.e.m. Symbols indicate individual donors (n = 3). Statistical significance was assessed using two-way ANOVA and post-hoc Tukey test. (NS, P = 0.961 sPSMA vs aPSMA). Right: evaluation of mCherry expression (total red area per μm²) by activated anti-PSMA tumour-cell reactive CAR-T cells. Values for the IncuCyte assay are the mean ± s.e.m. for n = 3 human T-cell donors. Statistical significance was assessed by two-way ANOVA and post-hoc Tukey test. (*P = 0.0182 sPSMA vs aPSMA). **g**, Flow cytometric analysis to evaluate % mCherry (left) and mCherry MFI (right) background expression levels in non-activated CAR-T cells. **h**, Flow cytometric analysis to evaluate % mCherry (left) and mCherry MFI (right) expression by activated CAR-T cells upon 24 h co-culture with PSMA+ PC3-PIP tumour cells. **i**, Flow cytometric analysis to evaluate % mCherry (left) and mCherry MFI (right) by CAR-T cells after 24 h PMA-Ionomycin stimulation. Bar graphs (**g–i**) show the mean ± s.e.m. Symbols indicate individual healthy T-cell donors (n = 3). Statistical significance was assessed by one-way ANOVA (**g** left ****P < 0.001 sPSMA vs aPSMA; **g** right ***P < 0.001 sPSMA vs aPSMA; **h** left NS, P = 0.1699 sPSMA vs aPSMA; **h** right ****P < 0.001 sPSMA vs aPSMA; **i** left NS, P = 0.1492 sPSMA vs aPSMA; **i** right ***P < 0.001 sPSMA vs aPSMA). a, antisense; s, sense; mC, mCherry.

PMA-Iono activation we observed similar %mCherry expression for both CAR constructs (Extended Data Fig. 5b, right).

For the first in vivo study, NSG mice were inoculated with $5 \times 10^6$ PSMA⁺ PC3-PIP tumour cells and treated on day 5 by peritumoral transfer of $5 \times 10^6$ 4G CAR- or UTD-T cells (Extended Data Fig. 5c). As expected, the 4G anti-PSMA CAR-T cells, but neither the 4G anti-CD19 CAR- nor the UTD-T cells, were able to control tumour growth (Extended Data Fig. 5d). In addition, luciferase activity upon luciferin injection in mice was only observed for the tumour-infiltrating 4G anti-PSMA CAR-T cells (Extended Data Fig. 5e,f). Subsequently, we repeated the in vivo study but further compared next-generation anti-PSMA CAR-T cells generated with antisense vs sense vectors (Fig. 5a,b). We observed no

significant differences in tumour control for antisense vs sense lentiviral vector-generated CAR-T cells (Fig. 5c), in line with our in vitro data for this very potent anti-PSMA CAR. In this study, we also sought to evaluate whether the use of Tax and NovB2 during lentivirus vector production (to increase titres) has any impact on CAR-T-cell function. We found that there was no significant difference in tumour control by anti-PSMA CAR-T cells generated with virus produced in the presence or absence of Tax and NovB2 (Fig. 5c). Importantly, however, we observed that luciferase activity levels of tumour-infiltrating CAR-T cells, as measured by luminescence imaging upon luciferin injection of the treated mice, were significantly higher for the antisense lentiviral vector-generated 4G CAR-T cells (Fig. 5d,e). This observation, which

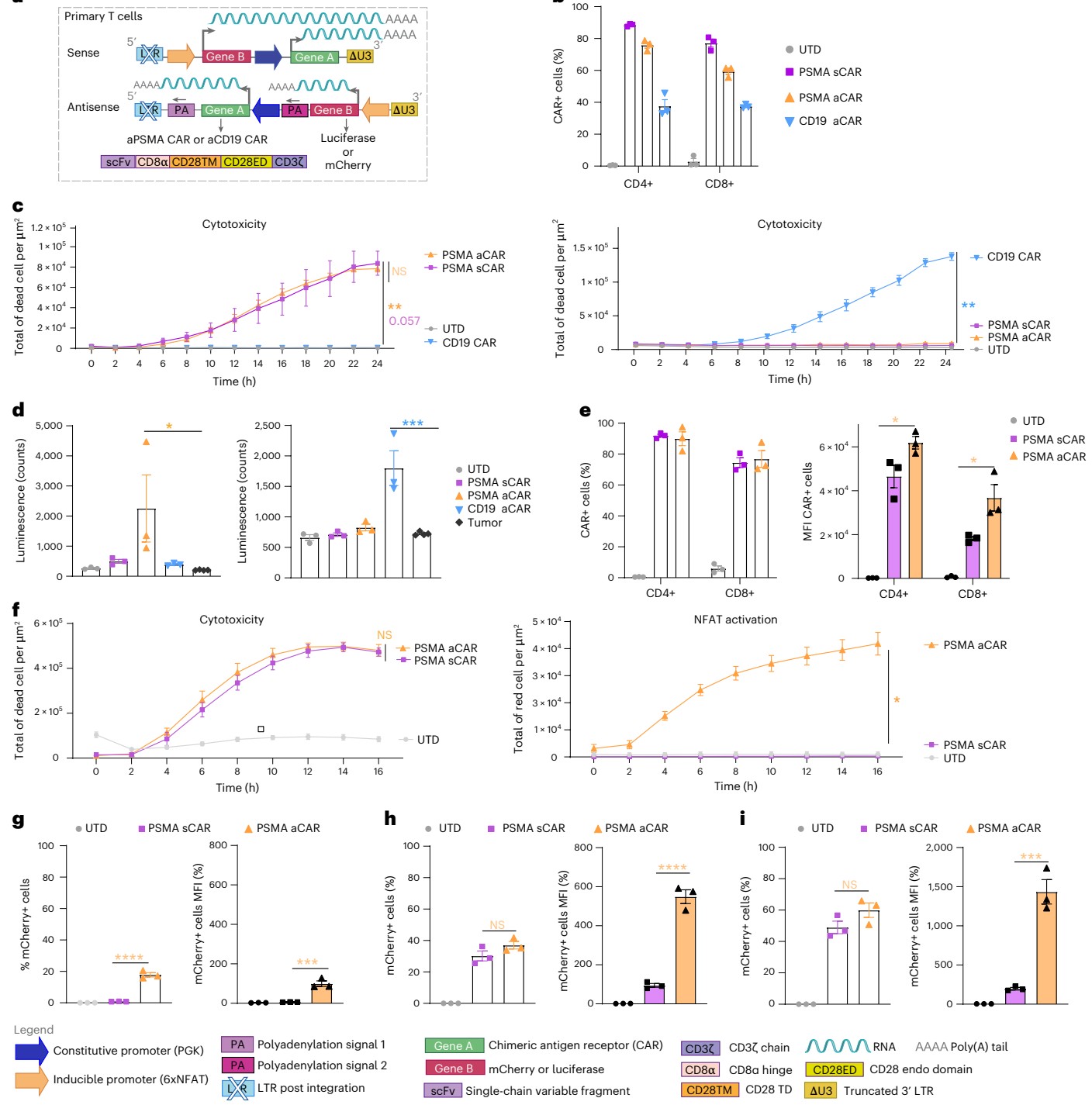

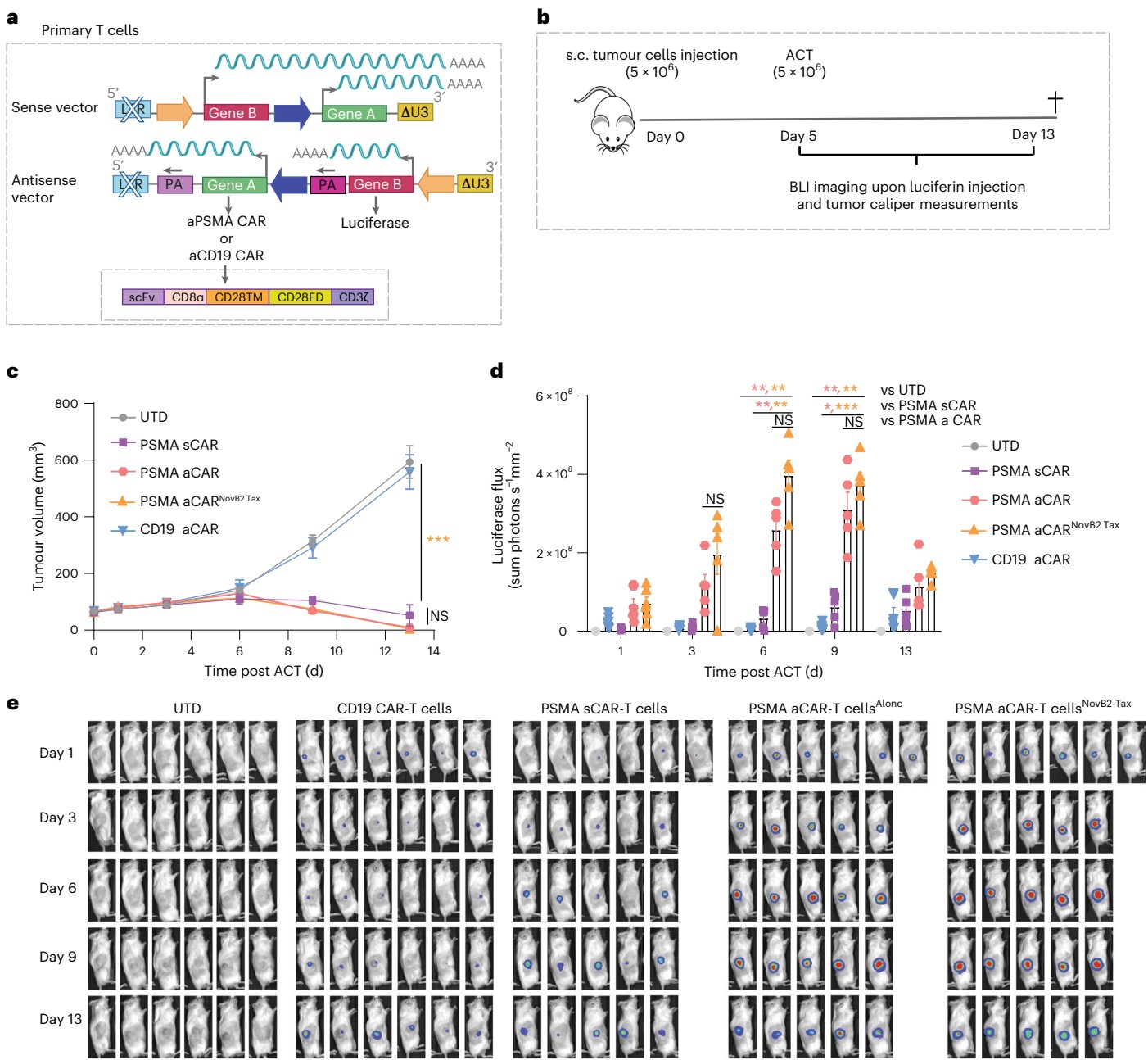

**Fig. 5 | In vivo testing reveals higher activation-induced expression levels of gene cargo by 4G CAR-T cells engineered with antisense vs sense lentiviral vectors. a,** Schematic of sense and antisense lentiviral vectors encoding anti-PSMA and anti-CD19 CARs under the PGK promoter and luciferase under 6xNFAT. **b,** Schematic of the in vivo study. **c,** Caliper tumour volume measurements over days. Values are the mean ± s.e.m. for $n$ = 6 mice per group. Statistical significance was determined by two-way ANOVA; ***$P$ < 0.001 aPSMA vs UTD at endpoint; NS $P$ = 0.78 aPSMA vs sPSMA. **d,** Luciferase flux as measured by bioluminescence imaging upon luciferin injection for all the experimental groups. Data are represented as mean ± s.e.m. for $n$ = 6 mice per group. Statistical significance was assessed using two-way ANOVA and post-hoc Tukey test; ****$P$ < 0.0001; Day 3: NS $P$ = 0.67 aPSMA vs aPSMA Tax-NovB2; Day 6 NS $P$ = 0.13 aPSMA vs aPSMA Tax-NovB2; **$P$ = 0.006 sPSMA vs aPSMA; **$P$ = 0.002 sPSMA vs aPSMA Tax-NovB2; **$P$ = 0.006 UTD vs aPSMA; **$P$ = 0.002 UTD vs aPSMA Tax-NovB2; Day 9 NS $P$ = 0.78 aPSMA vs aPSMA Tax-NovB2; *$P$ = 0.01 sPSMA vs aPSMA; ***$P$ < 0.001 sPSMA vs aPSMA Tax-NovB2; **$P$ = 0.009 UTD vs aPSMA; **$P$ = 0.002 UTD vs aPSMA Tax-NovB2. **e,** Representative images of luciferase activity of the transferred tumour-infiltrating 4G CAR-T cells over days upon luciferin injection of mice.

correspond with our in vitro findings (Fig. 4), is presumably due to a lack of transcriptional interference in the engineered T cells as occurs upon the use of dual sense lentiviral vectors.

Finally, we evaluated the next-generation CAR-T cells in vivo against a CD19⁺ tumour model. Briefly, mice were inoculated with 10 × 10⁶ Bjab tumour cells and on day 7 were treated by peritumoural transfer of 5 × 10⁶ antisense lentiviral vector-generated 4G CAR-T cells or UTD-T cells (Extended Data Fig. 5g). As expected, the anti-CD19 CAR-, but neither the anti-PSMA-CAR- nor the UTD-T cells, were able to control tumour growth. We also observed no significant differences in tumour control (Extended Data Fig. 5h) or in NFAT-driven luciferase activity (Extended Data Fig. 5i,j) for 4G anti-CD19 CAR-T cells generated

with virus produced in the presence or absence of Tax and NovB2. We further showed that the use of NovB2 and Tax during lentivirus vector production (Extended Data Fig. 6a) had no impact on transduction efficiency (Extended Data Fig. 6b), the cytolytic capacity of CAR-T cells against target cells (Extended Data Fig. 6c), the levels of inducibly expressed gene cargo upon CAR-T cell co-culture with target cells (Extended Data Figs 6d,f) or tumour control by anti-CD19 CAR-T cells (Extended Data Fig. 6g,h).

### Development of culture conditions suitable for clinical-grade lentivirus vector production

HTLV-Tax has been reported to act on several signalling pathways, among them NF-κB[71]. Although no Tax protein is expected in the lentiviral particle preparation following ultracentrifugation, its tumourigenic potential[65] may raise regulatory concerns for clinical-grade production of lentivirus vector. We thus sought to identify a suitable alternative. As previously mentioned, the CMV promoter and enhancer comprises four NF-κB consensus binding sites, and TNFα, IL-1β, camptothecin and phorbol ester (PMA) have all been shown to efficiently activate NF-κB in a dose-dependent manner[72]. To validate this effect in a simple manner, we transiently transfected HEK293T cells with a suboptimal concentration of pcDNA-EGFP which harbours a CMV promoter and treated the cells with the different compounds. At 48 h post-transfection, we observed an increase in both the percentage and MFI of cells expressing EGFP upon TNFα exposure (Extended Data Fig. 7a). Encouraged by this observation, we next tested the use of TNFα in the context of sense-orientation single gene cassette (Extended Data Fig. 7b) lentivirus vector production in HEK293T cells and observed an important increase in viral titre, percentage and MFI of EGFP+ cells (Extended Data Fig. 7c), presumably due to the effect of TNFα not only on the transfer vector but also on the envelope and packaging vectors which comprise CMV promoters. Of note, this NF-κB-mediated strategy can in principle be applied to enhance the production and hence lower the costs of any viral vector comprising NF-κB consensus binding sites in promoter/enhancer regions.

### Evaluation of clinical-grade protocol in the context of 'difficult to produce' lentivirus vectors

Along with the development of tumour-redirected T cells that co-express additional molecules or receptors, gene-downregulation strategies[30] can also be employed to potentiate their function. However, transfer vectors encoding shRNA which comprise stem-loop structures are associated with low viral titres due to Dicer processing. Hence, to further validate the use of TNFα and NovB2 to augment viral titres, we developed different transfer vectors comprising a short miR-based shRNA hairpin[73–75] (miR-based shRNA). Notably, NovB2 has been previously shown to increase the titre of such vectors due

to specific inhibition of the canonical activity of Dicer isoforms in processing microRNAs[76].

We began by expressing the miR-based shRNA under the constitutive U6 promoter with EGFP expressed downstream under the PGK promoter (Fig. 6a). Indeed, because the termination of transcription from polymerase III promoters comprises 5 thymidine residues, the vector was built in a dual sense orientation; there is no transcriptional interference to reach a PA site and hence no need to invert the gene cassette. Upon titration of viral supernatant produced in the presence of NovB2, TNFα or both, we observed an important gain in transduction efficiency as measured by percentage of EGFP+ cells (Fig. 6b), lentiviral titre (Fig. 6c) and relative expression level of EGFP per cell (MFI) (Fig. 6d).

Encouraged by these results, we subsequently built a sense vector comprising a miR-based shRNA under the U6 promoter targeting a therapeutically relevant target, Hematopoietic Progenitor Kinase 1 (Hpk1), a negative regulator of TCR signalling[77], also known as Mitogen-Activated Protein Kinase 1 (Map4k1). The miR-based shRNAs were followed by truncated human nerve growth factor receptor (tNGFR)[78] and the HLA-A2/NY-ESO-1$_{157-165}$ restricted TCR[69], both expressed under the PGK promoter and separated by a T2A element (Fig. 6e). Jurkat cells transduced with this construct showed an efficient knockdown of HPK1 (over 90% reduction by miR-based shRNA 'A') (Fig. 6f). We then transduced primary T cells and observed 85% and ~70% transduction efficiency of primary CD4+ and CD8+ T cells, respectively, as measured by HLA-A2/NY-ESO-1$_{157-165}$ tetramer staining (Fig. 6g). Efficient transduction was accompanied by strong HPK1 knockdown, similar to the levels observed in Jurkat cells (Fig. 6h). We subsequently evaluated the in vitro function of the TCR-T cells +/− HPK1 knockdown by miR-based shRNA upon co-culture with the A2+/NY+ target cell lines Me275 and A375, as well as the A2+/NY− cell line Na8 as a negative control. Others have previously demonstrated that pharmacological inhibition or full gene knock-out of HPK1 in CD8+ T cells can improve their effector function and ability to control tumours[77,79]. However, we did not observe significant differences in target-cell killing (Fig. 6i) or in IFNγ release (Fig. 6j) for the HPK1 knockdown TCR-T cells (HPK1 'A' and 'B') vs the control (CTRL) TCR-T cells comprising a scrambled miR-based shRNA, but we did observe higher proliferative capacity for the HPK1 'A' knockdown CD8+ T cells (Fig. 6k). Whether these differences are due to the use of miR-based shRNA to knockdown HPK1 or the in vitro conditions used in our experiments is unknown, but is beyond the scope of our study.

### Evaluation of clinical-grade lentivirus vector production protocol for antisense transfer vectors

The use of TNFα in combination with NovB2 was next tested in the context of the antisense configuration transfer vector encoding mCherry under 6xNFAT and EGFP under PGK (Fig. 7a, left). Similar to when Tax

**Fig. 6 | Optimized lentivirus vector production protocol yields high titres in the context of transfer vectors encoding miR-based shRNA. a**, Schematic of sense lentiviral transfer vector encoding a chimeric CMV promoter and enhancer at the 5′ LTR to allow enhanced replication in the presence of TNFα and EGFP. **b**, Transduction of Jurkat cells with decreasing volumes of lentivirus vector supernatant produced in the presence or absence of TNFα and NovB2, and flow cytometric evaluation (on day 5) of % EGFP expression. Bar graph represents the mean of 5 independent experiments. **c**, Viral titres (TU ml$^{-1}$). **d**, Representative histograms of EGFP expression by Jurkat cells transduced with 100 μl of lentivirus vector supernatant. **e**, Schematic of sense lentiviral transfer vector encoding miR-based shRNA targeting HPK1 (shRNA A and shRNA B) or scramble control (shRNA CTRL) under the U6 promoter, as well as truncated nerve growth factor receptor (tNGFR) and a TCR, both under the PGK promoter and separated by T2A sequences. **f**, Western blot analysis to evaluate HPK1 downregulation in Jurkat cells (technical replicates shown), together with β-actin control. **g**, Transduction efficiency of primary human CD4+ and CD8+ T cells with lentivirus vector supernatant produced in the presence of TNFα and NovB2. At 5 d

post-transduction, the T cells were stained with HLA-A2/NY-ESO-1$_{157-165}$ tetramer and analysed by flow cytometry. Bar graph represents the mean ± s.e.m. of n = 3 human T-cell donors. **h**, Western blot analysis to evaluate HPK1 downregulation, together with β-actin control blot for n = 3 human donors (HD) (Source Data for Fig. 6). **i**, Tumour-cell killing assay for Nuclei red A2+/NY+ targets Me275 and A375, and Nuclei red and A2+/NY− cell line Na8, by TCR-T cells with miR-based shRNA knockdown of HPK1, TCR-T cells comprising a scrambled miR-based shRNA (CTRL) and UTD-T cells, as measured by the IncuCyte instrument as a loss in red area over time. Shown are mean ± s.e.m. for n = 3 independent T-cell donors. **j**, IFNγ release as measured by ELISA upon 24 h co-culture of TCR-T cells with miR-based shRNA knockdown of HPK1 T cells, CTRL- or UTD-T cells with A2+/NY+ targets Me275, A375 and Saos-2, and A2+/NY− cell line Na8. Bar graphs represent the mean ± s.e.m. for n = 3 human T-cell donors. **k**, Percentage of CTV negative cells (cells that have undergone proliferation) upon tumour stimulation. Shown are mean ± s.e.m. for n = 3 healthy T-cell donors. Statistical significance was assessed using two-way ANOVA, *P = 0.0209 HPK1 vs UTD CD8+.

was used, a gain in viral titre was observed in the presence of TNFα alone, but titres were even higher if NovB2 was combined with TNFα (Fig. 7a, middle and right).

It is well known that the use of vectors comprising U6-driven shRNAs can be toxic to transfected cells[80–82], and polymerase III promoters do not allow for inducible expression of genes of interest. Hence, to overcome this obstacle we next built an antisense vector comprising an miR-based shRNA under 6xNFAT and EGFP under PGK (Fig. 7b, left), and produced lentivirus vectors using our optimized clinical-grade production protocol. We observed an important gain in viral titre in

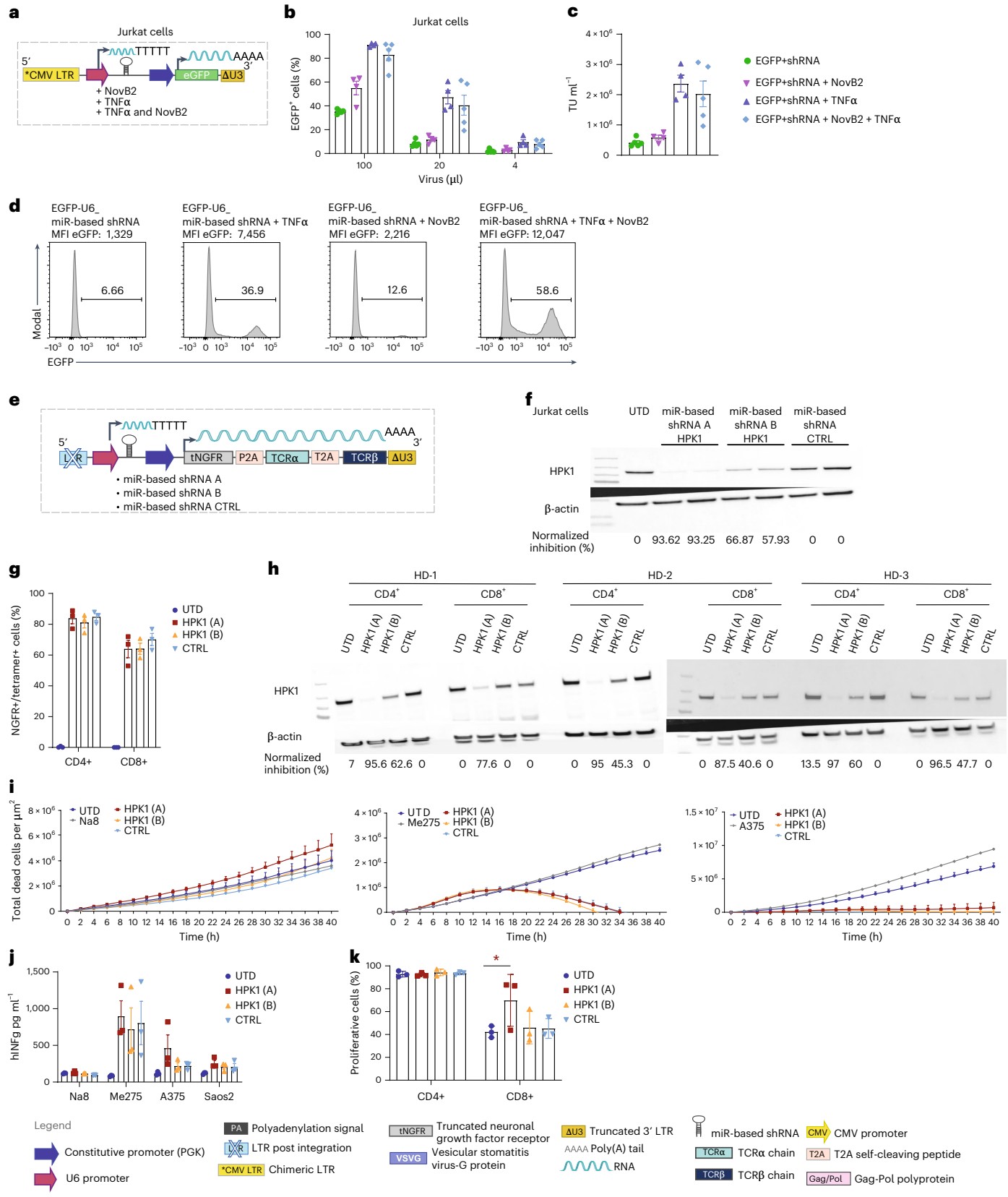

the presence of NovB2 alone or combined with TNFα (Fig. 7b, middle and right).

We further evaluated an inverted configuration vector comprising the anti-PSMA CAR and miR-based shRNA 'A' targeting HPK1 under 6xNFAT in primary human T cells (Fig. 7c, left). Upon transduction with lentivirus vector produced in the presence of NovB2 and TNFα, we reached approximately 90% and about 60% CAR expression by CD4[+] and CD8[+] T cells, respectively (Fig. 7c, middle). Moreover, upon 6 h CAR-T-cell triggering with plate-coated anti-F(ab), we achieved over 90% HPK1 knockdown (Fig. 7c, right).

Finally, for the dual antisense vector, we cloned an miR-based shRNA targeting the TCR-alpha chain under an alternative constitutive Polymerase II promoter, SFFV (silencing prone spleen focus forming virus), and EGFP under PGK (Fig. 7d, left). This is a strategy that can be used to abrogate TCR chain mispairing upon engineering of T cells for ACT with an exogenous TCR [25]. Transduced Jurkat cells demonstrated efficient knockdown of the TCR-alpha chain with our dual antisense vector as measured by cell-surface staining with a pan-anti-TCR antibody (Fig. 7d, bottom right).

Thus, in summary we have demonstrated that the use of TNFα during virus production when using antisense (or sense) transfer vectors in which the RSV-based promoter and enhancer at the 5' LTR is replaced with the complete CMV promoter and enhancer (which comprises 4 consensus NF-κB binding motifs[67]) can substantially increase titres. It is likely that the TNFα, in addition to favouring transcription of the transfer vector, also promotes replication of the packaging and envelope vectors. Moreover, the presence of TNFα in the culture media can synergize with NovB2, a protein that can abrogate Dicer-mediated dsRNA antiviral response generated during virus production in HEK293T cells. In addition, the protocol, which is feasible for the production of clinical-grade virus at reduced costs, can be used to generate high titres of 'difficult to produce' lentivirus vector, such as ones encoding miR-based shRNA. Indeed, NovB2 may further abrogate Dicer-mediated processing of such hairpin structures.

## Discussion

In recent years, rational TCR- and CAR-T-cell co-engineering strategies have been under extensive investigation to improve responses against solid tumours, either by directly enhancing the intrinsic fitness and function of the T cells themselves or/and by TME reprogramming[8,30]. In addition to barriers in the TME, the clinical success of T-cell therapy against solid tumours is constrained by adverse patient reactions such as on-target but off-tumour toxicity[83], as well as cytokine release syndrome by CAR-T cells[84] and unexpected cross-reactivity by TCR-T cells against vital organs[85]. Hence, important research efforts are also being undertaken in the development of ON, OFF and STOP switches[68,86–88] along with gene-modification strategies[89] and optimized vectors[41] to allow tighter control of the biological activities of engineered T cells post-infusion[30].

Although emerging gene-modification strategies such as Crispr/Cas9 hold tremendous potential for the development of next-generation TCR- and CAR-T cells[19], in particular for gene knock-outs[1] but also gene knock-ins as approaches are developed to increase efficiencies[90,91], the (1) strong safety record of lentiviral vectors coupled with (2) enhanced manufacturing protocols[28] and (3) the high transduction efficiencies that can readily be achieved make lentivirus vectors an important clinical tool. Indeed, lentivirus vectors will probably be used for years to come in the clinic, likely also in combination with Crispr/Cas9[19] and other gene-engineering techniques. Hence, further optimization of lentiviral vectors, virus production methods and transduction strategies are warranted[23].

Here we have developed an antisense transfer vector allowing efficient constitutive expression of a tumour-directed TCR or CAR and independent co-expression of gene cargo. While we have used the activation-inducible promoter 6xNFAT to express various gene cargoes including IL-2 and miR-based shRNAs to knockdown genes of interest, it is also feasible to employ promoters that respond to environmental cues including hypoxia[92]. Such an approach may be useful, for example, for co-expression of chemokines that can generate a gradient to attract additional lymphocytes into the tumour bed. The development of drug-inducible promoters[93], such as the tetracycline-controlled ON system (Tet-ON, of bacterial origin)[94], comprising non-immunogenic components suitable for the clinic and allowing sufficient expression levels of the target molecule(s) of interest for therapeutic efficacy, would be of great benefit for tighter and safer control of next-generation TCR- and CAR-T cells and other cellular therapies.

In our study, side-by-side evaluation with comparative dual forward and bidirectional vectors revealed transcriptional interference for the former and leakiness of the inducible promoter for the latter configuration. However, we showed that primary human T cells could be efficiently engineered with lentivirus vector comprising a dual antisense transfer vector encoding a constitutively expressed CAR or TCR and inducible gene cargo without such problems. Moreover, next-generation TCR- and CAR-T cells engineered with the dual antisense lentiviral constructs were validated for functionality both in vitro and in vivo in the context of solid tumour-bearing mice.

While the antisense transfer vector design was limiting to virus production, evidently because of convergent transcription in HEK293T cells, we developed a robust protocol to restore titres. First, we showed that the presence of the RNA interference suppressor protein NovB2[61], previously demonstrated to inhibit isoforms of Dicer[63], could augment lentiviral titres. We subsequently sought to address the issue that transcriptional interference is limiting to the levels of the ssRNA viral genome available for packaging. We began by using the Tax protein[66] which, in addition to a variety of oncogenic properties, can act as a potent transactivator of CMV promoters as they harbour 4 NF-κB binding motifs[67]. Indeed, we replaced the RSV-based promoter and enhancer at the 5' LTR of the transfer vector with the complete CMV promoter and

---

**Fig. 7 | Optimized clinical-grade protocol for high-titre lentivirus vector production can be used in the context of antisense vectors encoding miR-based shRNA. a**, Left: schematic of antisense lentiviral transfer vector encoding EGFP under PGK and mCherry under 6xNFAT. Middle: transduction of Jurkat cells with titrated lentivirus vector supernatant produced in the presence or absence of TNFα in combination with NovB2; flow cytometric evaluation of % EGFP expression on day 5. Bar graphs represent the mean ± s.e.m. of 3 independent experiments. Right: viral titres (TU ml$^{-1}$). **b**, Left: schematic of dual antisense lentiviral transfer vector encoding EGFP under PGK and miR-based shRNA under 6xNFAT. Middle: transduction of Jurkat cells with titrated lentivirus vector supernatant produced in the presence or absence of TNFα or Tax in combination with NovB2; flow cytometric evaluation of % EGFP expression on day 5. Bar graphs represent the mean ± s.e.m. of 5 independent experiments. Right: viral titres (TU ml$^{-1}$). **c**, Left: schematic of antisense lentiviral transfer vector encoding an anti-PSMA-CAR under PGK and miRNA under 6xNFAT.

Middle: transduction efficiency of primary human CD4[+] and CD8[+] T cells with lentivirus vector supernatant produced in the presence of TNFα and NovB2. T cells were stained with fluorescenated anti-Fab Ab to evaluate cell-surface CAR expression on day 5 post-infection. Bar graphs represent the mean ± s.e.m. of n = 4 human T-cell donors. Right: western blot analysis showing specific downregulation of HPK1 upon 6 h stimulation with plate-coated anti-F(ab)$_2$, together with β-actin control blot of n = 2 human T-cell donors (Source Data for Fig. 7). **d**, Top left: schematic of antisense lentiviral transfer vector encoding EGFP under PGK and miR-based shRNA targeting TRAC, or control miR-based shRNA, under the constitutive promoter SFFV. Bottom left: representative dot plot of flow cytometric evaluation of % EGFP expression on day 5 and PAN-anti-TCR antibody staining to evaluate TCR knockdown. Top right: transduction of Jurkat cells with different amounts of lentivirus vector supernatant. Bar graphs represent the mean ± s.e.m. of EGFP[+] cells. Bottom right: the percentage of TCR[+] cells for 3 independent experiments.

enhancer, and showed that we could increase viral titres in the presence of Tax[65], and to a greater extent when combined with NovB2.

For potential clinical GMP-grade production of lentivirus vector, we sought a substitution for Tax. We demonstrated that the presence of TNFα (previously shown to efficiently act on NF-κB binding motifs in a dose-dependent manner[72]) in the culture supernatant also increased viral titres. Notably, the use of TNFα to increase viral titres may be applicable to other viruses produced from vectors comprising promoters with NF-κB binding motifs. Moreover, TNFα may be useful for increasing plasmid production (that is, comprising NF-κB binding motifs) in transfected cells.

Recently, an 'all in one' dual sense lentiviral vector system was described comprising inducible expression of a gene upstream of a constitutively expressed second gene. However, in line with previous work, our data suggest transcriptional interference for this design and consequently lower gene expression[56], presumably due to competition for the same PA site and the simultaneous occupancy of the DNA template. This lower expression may be limiting to the therapeutic efficacy of the cellular product, such as T cells gene-modified to secrete a decoy molecule targeting an immune checkpoint such as programmed death-protein 1 (PD-1)[95]. The enhanced expression of genes from our dual inverted vector is probably due both to a lack of transcriptional

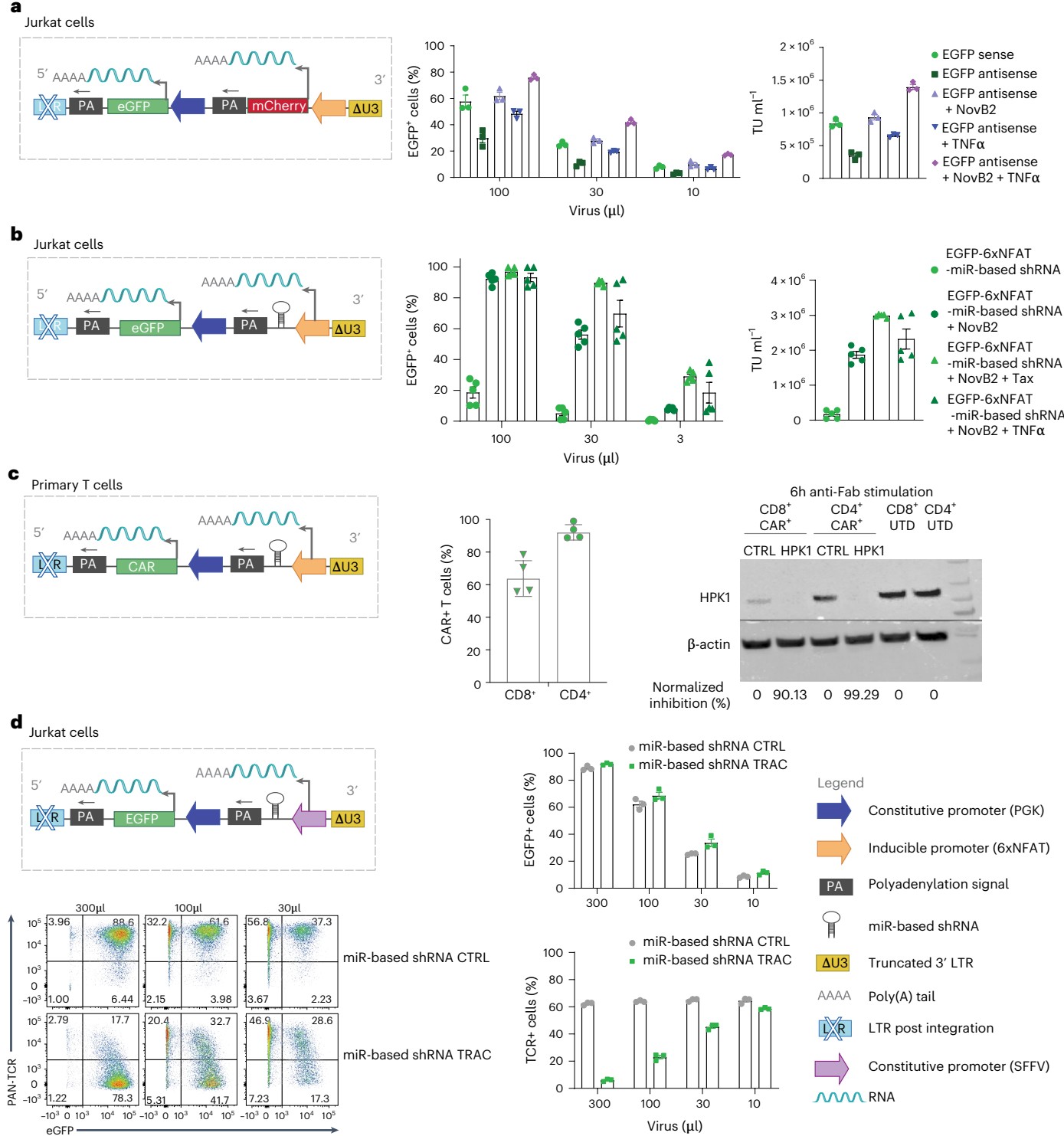

interference as well as the use of the potent BGH polyadenylation signal[58]. A full head-to-head comparison of the vector designs cannot be undertaken as it is not possible to include independent PA sites in the sense vector as this will abrogate virus production. Notably, it is important to evaluate potential 'leakiness' from vectors comprising inducible promoters. For example, tonic CAR signalling[70] can lead to gene expression under 6xNFAT in a target antigen-independent manner.

We further tested a bidirectional transfer vector design but observed expression of the inducible gene in non-activated cells. While it may be possible to abrogate leakiness by further buffering the two promoters, this will be limiting to the size of the genes that can subsequently be accommodated; beyond a genomic load of 10,000 bp, lentiviral vectors become increasingly inefficient[96,97]. We did not test a convergent design for the transfer vectors because we reasoned that there would be interference in gene expression in transduced T cells[42]. Moreover, a convergent design runs the risk of an unwanted IFN response in gene-modified T cells due to the generation of dsRNA.

Taken together, our work presents an improved dual antisense transfer vector and accompanying lentivirus vector production protocol enabling efficient transduction of primary human T cells with a constitutively expressed tumour-targeting receptor along with independent activation-inducible co-expression of gene cargo. We demonstrated functionality of the dual inverted vector encoding either a CAR or a TCR under PGK and various gene cargoes under 6xNFAT including IL-2 and miR-based shRNA targeting HPK1. We further demonstrated proof-of-principle for the use of our dual inverted vector for generating 4G CAR-T cells for ACT. We showed that the inducible gene cargo (luciferase) was expressed by T cells in tumours only if target antigen for the CARs was present. Notably, our overall approach is universal in that it can be applied to the engineering of other cell types, alternative polymerase II promoters and different engineering purposes in the context of other diseases. Importantly, our strategy can lower costs due to the use of a single vector and higher titres achieved, and it holds important promise towards effective and safety-enhanced next-generation cellular therapies reaching the clinic.

## Methods

### Cell lines and culture
The prostate carcinoma cell line PC3-PIP (PMSA+), PC3 engineered with human CD19+ cells, Bjab, Na8, Me275, A375, Saos-2, 293T human embryonic kidney (HEK293T) cells and Jurkat cells were cultured in RPMI-1640 medium supplemented with 10% heat-inactivated fetal bovine serum (FBS), 2 mmol l⁻¹ L-glutamine, 100 µg ml⁻¹ penicillin and 100 U ml⁻¹ streptomycin, at 37 °C in a 5% CO$_2$ atmosphere (Invitrogen, Life technologies). Na8, Me275, A375, Saos-2, 293T and Jurkat cell lines were purchased from the ATCC. The PC3-PIP and PC3 cell lines were kindly provided by Dr A. Rosato (University of Padau, Padova, Italy)[98]. Bjab was kindly provided by Dr C. Arber (University of Lausanne, Switzerland). The PC3 and PC3-PIP cells lentivirally transduced to enforce expression of CD19 (PC3-CD19+ and PC3-PIP CD19+) were kindly provided by Dr Y. Muller (University of Lausanne, Switzerland). The HEK293T cell line was used for lentivirus vector production.

### Vector construction
Second-generation CARs comprising the CD8α hinge, CD28 transmembrane (TM), CD28 endodomain (ED) and CD3-zeta ED were cloned into a 2G self-inactivating lentiviral expression vector pELNS under the PGK promoter. The HLA-A2/NY-ESO-1$_{157-165}$ restricted TCR was cloned in vector pRRL, with expression also driven by the PGK promoter. The anti-PSMA scFv derived from monoclonal antibody J591[99] and the anti-CD19 CAR scFv derived from monoclonal antibody FMC63[100] were used to confer tumour-antigen specificity. The HLA-A2/NY-ESO-1$_{157-165}$ restricted TCR has been previously described[69]. The (NFAT)$_6$ response elements-IL-2 minimal promoter, abbreviated as 6xNFAT, was used to evaluate inducible expression of different gene cargoes. Replacement

of the RSV promoter with the CMV promoter in the 5' LTR was used to enable TNFα in the culture supernatant to favour transcription of the ssRNA viral genome.

### Lentivirus vector production
For large-scale production: briefly, 24 h before transfection, 293T cells were seeded at 10 × 10⁶ cells in 30 ml medium in a T-150 tissue culture flask. All plasmid DNA was purified using the Endo-free Maxiprep kit (Invitrogen, Life Technologies). 293T cells were transfected with 7 µg pVSVG (VSV glycoprotein expression plasmid) or 7 µg pVSVG-T2A-NovB2, 18 µg of R874 (Rev and Gag/Pol expression plasmid) and 15 µg of pELNS or pCRRL transgene plasmid using a mix of Turbofect (Thermo Fisher) and Optimem media (Invitrogen, Life Technologies, 180 µl of Turbofect for 3 ml of Optimem). The cells were further transfected with a plasmid encoding the T-cell leukaemia virus 1, TAX protein, or the medium was further supplemented with TNFα at 10 ng ml⁻¹ working concentration. The viral supernatant was collected at 48 h post-transfection. Viral particles were concentrated by ultracentrifugation for 2 h at 24,000 $g$ and resuspended in 400 µl complete RPMI-1640 media, followed by immediate snap freezing on dry ice.

For small-scale production: briefly, 4–5 h before transfection, 293T cells were seeded at 1.25 × 10⁶ cells in 2 ml medium per well in a 6-well plate. 293T cells were transfected with 2.5 µg total DNA (divided as 0.282 µg pVSVG or pVSVG-T2A-NovB2, 0.846 µg R874, and 1.125 µg pELNS or pCRRL transgene plasmid), using a mix of Lipofectamine 2000 (Invitrogen) and Optimem media (Invitrogen, Life Technologies) according to the manufacturer's instructions. The cells were further transfected with a plasmid encoding the T-cell leukaemia virus 1, TAX protein, or the medium was further supplemented with TNFα at 10 ng ml⁻¹. The viral supernatant was collected at 48 h post-transfection and supernatant was used directly.

### Jurkat cell transduction for viral titration
Jurkat cells were suspended at 1 × 10⁵ cells per ml and seeded into 24-well plates at 1 ml per well. Different volumes of viral supernatant were used for transduction, as indicated, ranging from 300 µl down to 3 µl. Cell media were refreshed after incubation for 24 h at 37 °C. Viral titres (transducing units per ml (TU ml⁻¹)) were calculated as follows: ((total number of cells/100) × percentage of transduced cells) × dilution of the virus supernatant.

### Primary human T-cell purification, activation, transduction and expansion
Primary human T cells were isolated from the peripheral blood mononuclear cells (PBMCs) of healthy donors (HDs; prepared as buffycoats) collected with informed consent by the blood bank. Total PBMCs were obtained via Lymphoprep (Axonlab) separation solution by a standard protocol of centrifugation. CD4+ and CD8+ T cells were isolated by negative selection using magnetic beads following the manufacturer's protocol (easySEP, Stem Cell Technology). Purified CD4+ and CD8+ T cells were cultured separately in RPMI-1640 with Glutamax, supplemented with 10% heat-inactivated FBS, 100 U ml⁻¹ penicillin, 100 µg ml⁻¹ streptomycin sulfate, and stimulated with anti-CD3 and anti-CD28 monoclonal antibody (mAb)-coated-beads (Invitrogen, Life Technologies) at a 1:2 ratio of T cells to beads. T cells were transduced with lentivirus vector particles at 18–22 h post-activation. Human recombinant interleukin-2 (h-IL-2; Glaxo) was replenished every other day for a concentration of 50 IU ml⁻¹ until 5 d post-stimulation (day +5). At day +5, magnetic beads were removed, and h-IL-7 and h-IL-15 (Miltenyi Biotec) were added to the cultures at 10 ng ml⁻¹. A cell density of 0.5–1 × 10⁶ cells per ml was maintained for expansion. Rested engineered T cells were adjusted for equivalent transgene expression before all functional assays; the more efficiently transduced samples were diluted with appropriate numbers of UTD-T cells.

## Cytotoxicity assays

Cytotoxicity assays were performed using the IncuCyte Instrument (Essen Bioscience). Briefly, $1.25 \times 10^4$ target cells were seeded in flat-bottom 96-well plates (Costar, Vitaris). Four hours later, rested T cells (no cytokine for 48 h) were washed and seeded at $2.5 \times 10^4$ cells per well, at a 2:1 effector:target (E:T) ratio in complete media. No exogenous cytokines were added during the co-culture period of the assay. CytotoxRed or Caspase-3/7 green reagent (Essen Bioscience) was added at a final concentration of 125 nM in a total volume of 200 µl. Internal experimental negative controls were included in all assays, including co-incubation of UTD-T cells and tumour cells, as well as tumour cells alone, to monitor tumour cell death over time. As a positive control, tumour cells alone were treated with 1% triton solution to evaluate maximal killing in the assay. In some assays (as indicated in the figure legends), freshly generated nuclei red and nuclei green engineered tumour cells were used. The nuclei red/green target cells were generated with IncuCyte NucLight Lentivirus (Essen Bioscience) for nuclear-restricted expression of tagGFP2 (green fluorescent protein) and mKate2 (red fluorescent protein), according to the manufacturer's instructions. Activation of co-engineered TCR-T and CAR-T cells upon specific antigen stimulation was assessed by mCherry IncuCyte quantification over time. Images of total red area per well and green area per well were collected every 2 h of the co-culture. The total red area per well and green area per well were obtained using the analysis protocol provided by Essen Bioscience. Data were normalized by subtracting the background fluorescence observed at time 0 (that is, before any cell killing by CAR-T cells) from all further timepoints. Data are expressed as mean ± s.e.m. of different HDs .

## Cell staining and flow cytometric analysis

To evaluate CAR cell-surface expression, transduced cells were stained with fluorescenated anti-human F(ab')mAb (BD Biosciences). To evaluate TCR cell-surface expression, transduced cells were stained with fluorescenated HLA-A2/NY-ESO-$1_{157-165}$ tetramer produced in-house. Aqua live Dye BV510 and near-infrared fluorescent reactive dye (APC Cy-7) were used to assess viability (Invitrogen, Life Technologies). To evaluate mCherry induction upon stimulation, T cells were stained with near-infrared fluorescent reactive dye (APC Cy-7) (Invitrogen, Life Technologies). Acquisition and analysis were performed using a BD FACS LRSII flow cytometer and FACS DIVA software (BD Biosciences).

## Immunoblotting

Cells were lysed in RIPA buffer supplemented with Halt phosphate/protease inhibitors (Thermo Fisher) and boiled at 97 °C for 10 min with Bolt LDS sample buffer and reducing agent (Thermo Fisher). Protein samples (10 µg) were separated by SDS−PAGE and transferred to PVDF membranes using the iBlot2 system (Thermo Fisher). Antibody staining of the molecules of interest was carried out according to the manufacturer's instructions. Rabbit monoclonal antibody (EP630Y) specific to MAP4K1/HPK1 antibody (ab33910) was purchased from Abcam and anti-β-actin (sc-47778) from Santa Cruz. Images were acquired with a western blot imager (Fusion, Vilber Lourmat), and protein levels were quantified using the ImageJ software by analysing pixel intensity of the bands. Total HPK1 level was calculated by dividing its signal to the β-actin signal.

## Mouse strain and in vivo experimentation

NOD scid gamma (NSG) male mice were bred and housed in a specific and opportunistic pathogen-free (SOPF) animal facility at the University of Lausanne (Epalinges, Switzerland). All in vivo experiments were conducted in accordance with and approval from the Service of Consumer and Veterinary Affairs (SCAV) of the Canton of Vaud. All cages housed 5 mice in an enriched environment providing free access to food and water. Mice were monitored at least every other day for signs of distress during experimentation and euthanized at endpoint by carbon dioxide overdose.

## Subcutaneous tumour model and adoptive T-cell transfer

NSG male mice aged 8−12 weeks were subcutaneously injected with $5 \times 10^6$ PC3-PIP (or PC3-CD19$^+$) tumour cells or $10 \times 10^6$ Bjab. Once tumour was palpable (day 5 for PC3 and day 7 for Bjab), the mice were treated by peritumoural injection of $5 \times 10^6$ UTD or CAR-T cells. Tumour volume was assessed every other day by caliper measurement. Tumour volumes were calculated using the formula $V = 1/2(\text{length} \times \text{width}^2)$, where length is the greatest longitudinal diameter and width is the greatest transverse diameter determined via caliper measurement.

## In vitro bioluminescence assay to evaluate inducible gene cargo expression levels for sense vs antisense lentiviral vectors

To evaluate gene-cargo expression levels for CAR- or TCR-T cells transduced with sense vs antisense lentiviral vectors containing luciferase as the inducible gene cargo under 6xNFAT, $2.5 \times 10^4$ UTD and transduced T cells were co-cultured with target tumour cells at 1:1 E:T ratio for 24 h in 96-well plates. The following day, the culture media were washed away and 10 µl per well of opportunely diluted Reporter Lysis 5X buffer (Promega) was added and the cells resuspended. Luciferin (50 µl per well) (PerkinElmer) was then added and cell lysates were transferred into white 96-well white optiplates (PerkinElmer) for bioluminescence acquisition. Luciferase activity was measured by total counts acquired using the HIDEX sense 425-301i plate reader and software (Hidex).

## Proliferation assay

To assess the proliferative capacity of A2/NY-specific TCR-T cells co-expressing an miR-based shRNA, both transduced and UTD-T cells ($n = 3$ donors) were stained with CTV (Invitrogen, Life Technologies) according to the manufacturer's instructions. Cells were then stimulated for 96 h with anti-CD3 and anti-CD28 monoclonal antibody (mAb)-coated-beads (Invitrogen, Life Technologies) at a 2:1 ratio of beads:T cells, or with A2$^+$/NY$^+$ tumour cells lines (Me275, A375 and Saos-2) and an A2$^+$/NY$^-$ cell line (Na8 cells) at an E:T ratio of 1:1.

## In vivo bioluminescence imaging using luciferase

Luciferase expression was evaluated in vivo from day 1 to day 11 post T-cell transfer. Mice were injected intraperitoneally with 150 mg kg$^{-1}$ d-luciferin (PerkinElmer) in 100 µl of PBS and transferred into an anaesthesia chamber induced by 3% mixture of isoflurane and 1.5% oxygen. Anaesthetized animals were imaged at 10−35 min post-luciferin injection using the In-Vivo Xtreme system (In-Vivo Xtrem, Bruker) reducing anaesthesia level to 1%. The photons emitted from the luciferase-expressing T cells were quantified using Molecular Imaging software (Bruker). A pseudocolour image representing the luminescence flux intensity was generated (violet and red colours refer to the least and the most intense flux, respectively) and then superimposed over the greyscale reference image. The luminescent region of interest was determined by drawing a gate and intensity of the signal was measured as total photon s$^{-1}$ mm$^{-2}$, which correlates proportionally with the expression of luciferase gene in transduced T cells. Mice were euthanized when the tumour volume reached 1,000 mm$^3$ according to the following formula $V = 1/2(\text{length Å-} \times \text{width}^2)$, or when they met euthanasia criteria (weight loss, signs of distress) in accordance with the Swiss Federal Veterinary Office and the Cantonal Veterinary Office guidelines.

## Statistical analysis

GraphPad Prism 9.0 software was used to determine statistically significant differences using one-way analysis of variance (ANOVA) followed by Tukey post-hoc correction for multiple comparisons (column groups, one variable tested). A two-way repeated measurement ANOVA followed by Tukey post-hoc correction was used for statistical analysis of tumour growth curves, in vitro cytotoxicity and mCherry induction analysis (two-variables analysis for multiple groups). Differences were

considered significant when \*$P < 0.05$, very significant when \*\*$P < 0.01$ and highly significant when \*\*\*$P < 0.001$.

**Reporting summary**

Further information on research design is available in the Nature Portfolio Reporting Summary linked to this article.

## Data availability

The main data supporting the findings of this study are available within the article and its Supplementary Information. All raw data generated during the study are available from the corresponding authors on request. Source data for the figures are provided with this paper.

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

## Acknowledgements

We thank members of the Flow Cytometry Platform and the Animal Care Facility of the Epalinges Campus of the University of Lausanne for their excellent support, and Dr J. A. Rath from the University of Lausanne for his advice on in vivo imaging acquisition and analysis. This work was generously supported by Ludwig Cancer Research, an Advanced European Research Council Grant to G.C. (1400206AdG-322875), the Prostate Cancer Foundation, the Biltema Foundation, Cancera and the Swiss National Science Foundation to M.I. (SNSF# 310030_204326). We also thank the Swiss Confederation for generously supporting the PhD salary of K.O. via the Swiss Government Excellence Scholarships programme.

## Author contributions

M.I. and G.C. directed the study. M.I. supervised the research. P.R. and G.M.P.G.A. conceived and planned experiments and performed them with M.T, K.O., E.C., M.S., S.A. and R.V.d.S. Data were analysed and interpreted by P.R., G.M.P.G.A., E.C., M.T., S.A., K.O. and M.I. The manuscript was written by G.M.P.G.A. and M.I., and finalized by M.I.

## Funding

## Competing interests

In the past 3 years, G.C. has received grants and research support or has been co-investigator in clinical trials by Bristol-Myers Squibb, Tigen Pharma, Iovance, F. Hoffmann La Roche AG and Boehringer Ingelheim. The Lausanne University Hospital (CHUV) has received honoraria for advisory services that G.C. has provided to Genentech, AstraZeneca AG and EVIR. G.C. has previously received royalties from the University of Pennsylvania for CAR-T-cell therapy licensed to Novartis and Tmunity Therapeutics. A provisional patent regarding the dual inverted vector and associated methodologies for increasing virus titres as described in this manuscript has been filed (2022 U.S. Provisional Patent Application No. 63/290,528 ANTISENSE TRANSFER VECTORS AND METHODS OF USE THEREOF) with M.I., G.C., P.R. and G.M.P.G.A. as co-inventors. The other authors declare no competing interests.

## Additional information

**Extended data** is available for this paper at https://doi.org/10.1038/s41551-023-01013-5.

**Correspondence and requests for materials** should be addressed to George Coukos or Melita Irving.

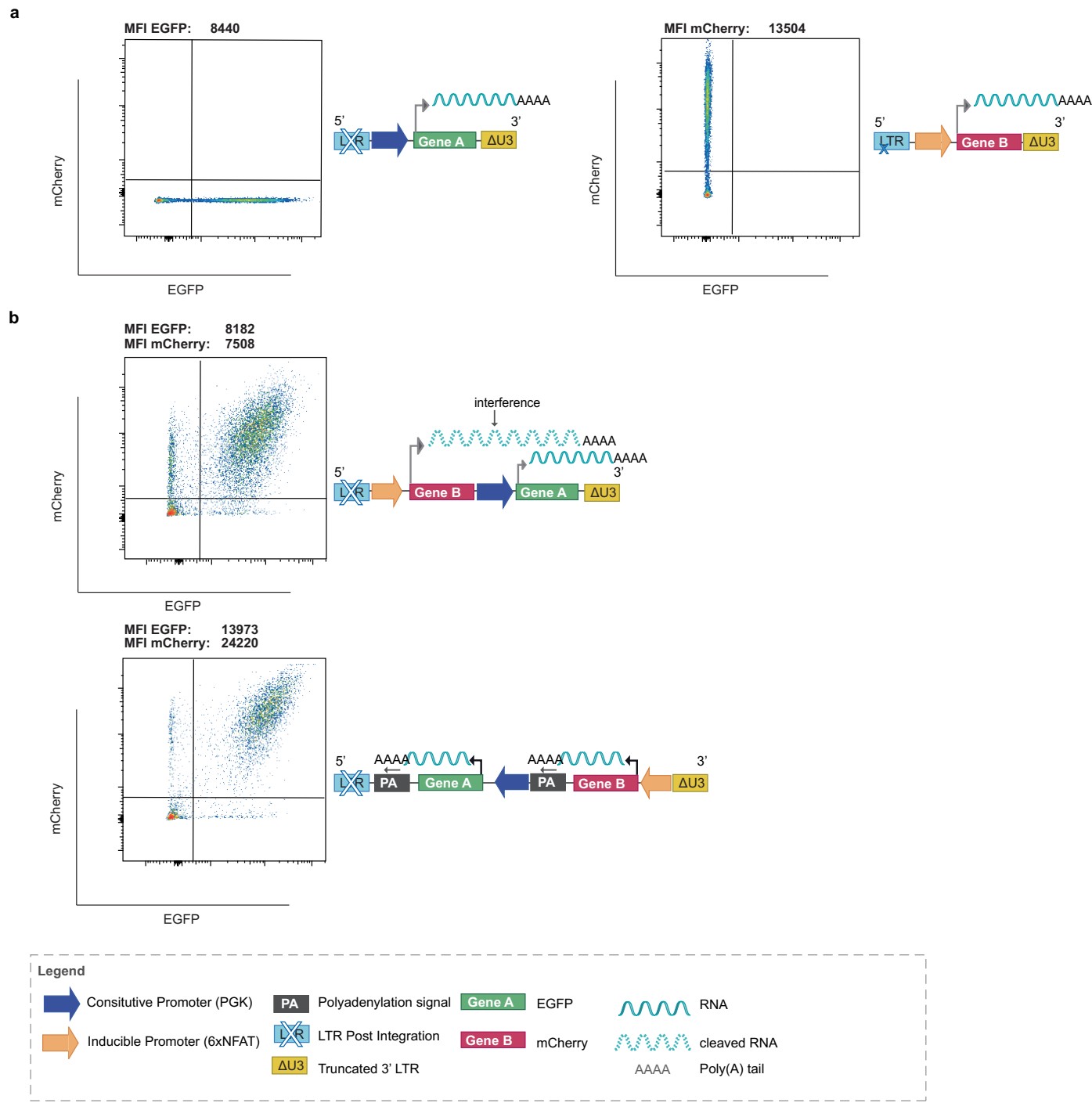

**Extended Data Fig. 1 | Antisense lentiviral vectors overcome the transcriptional interference that occurs for dual gene-cassette sense vectors. (a)** Representative flow cytometric analysis to evaluate expression levels (MFI) of EGFP (gene A) and mCherry (gene B) in activated Jurkat cells transduced with (**top**) single gene sense vectors in comparison to (**b**) sense (**top**) and antisense (**bottom**) dual gene cassette antisense vectors. Vector schematics are shown next to each plot. Plots are representative of three independent experiments each performed in replicate.

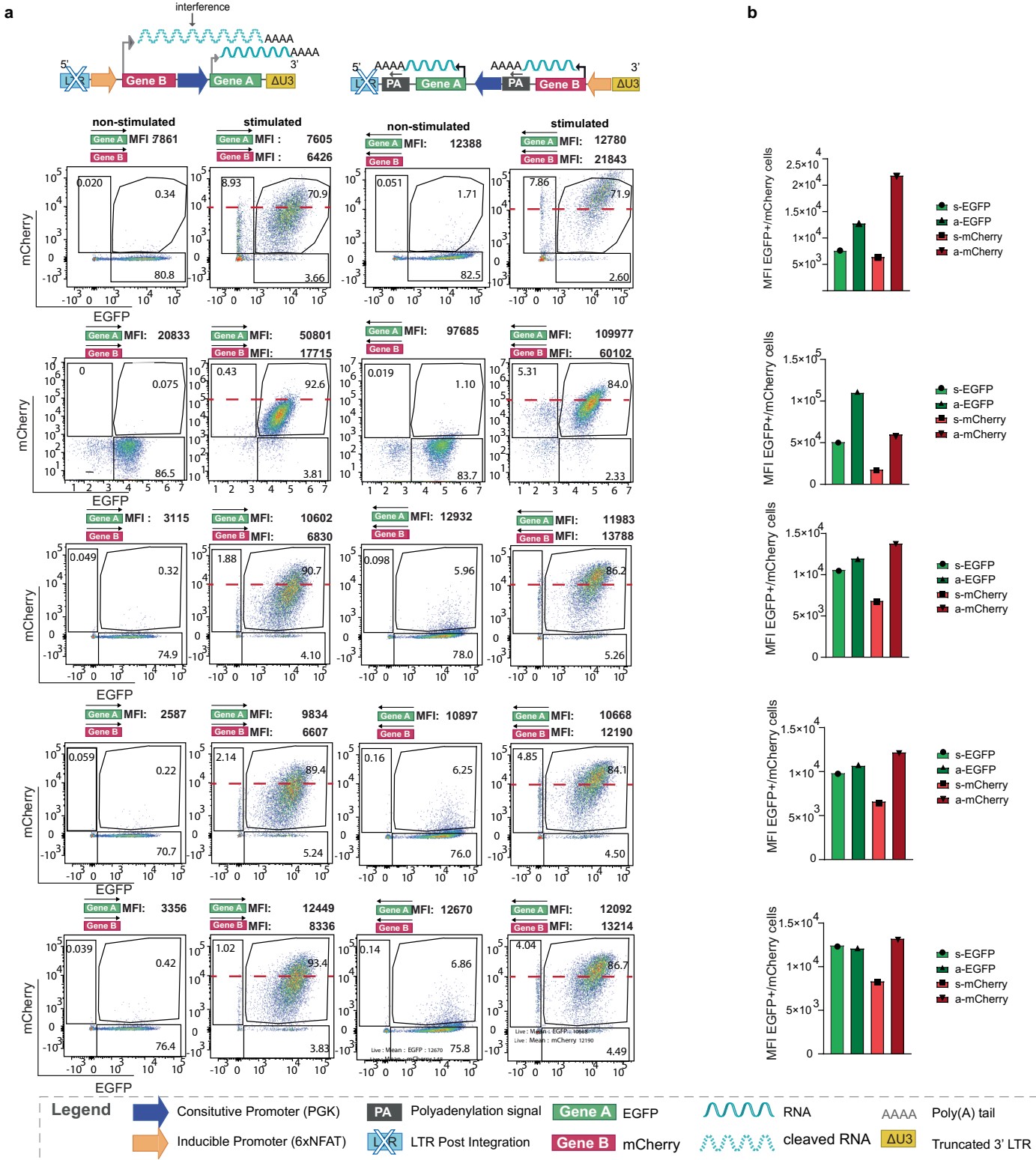

**Extended Data Fig. 2 | Higher gene expression levels in Jurkat cells transduced with dual antisense versus sense lentiviral vectors comprising inducible gene-cargo.** For all dual transfer constructs, EGFP (Gene A) expression is constitutively driven by the PGK promoter and mCherry (Gene B) by 6xNFAT as shown in the vector schematics on the top of FACS plots. **(a)** Representative flow cytometry dot plots for non-stimulated and stimulated Jurkat cells transduced with dual sense (left) versus antisense (right) orientation lentiviral vectors. Each FACS dot plot set corresponds to an independent experiment (total independent experiments = 5) **(b)** Bar graph representing the Mean Fluorescence Intensity (MFI) for EGFP and mCherry in stimulated Jurkat cells transduced with sense ('s') versus antisense ('a') constructs.

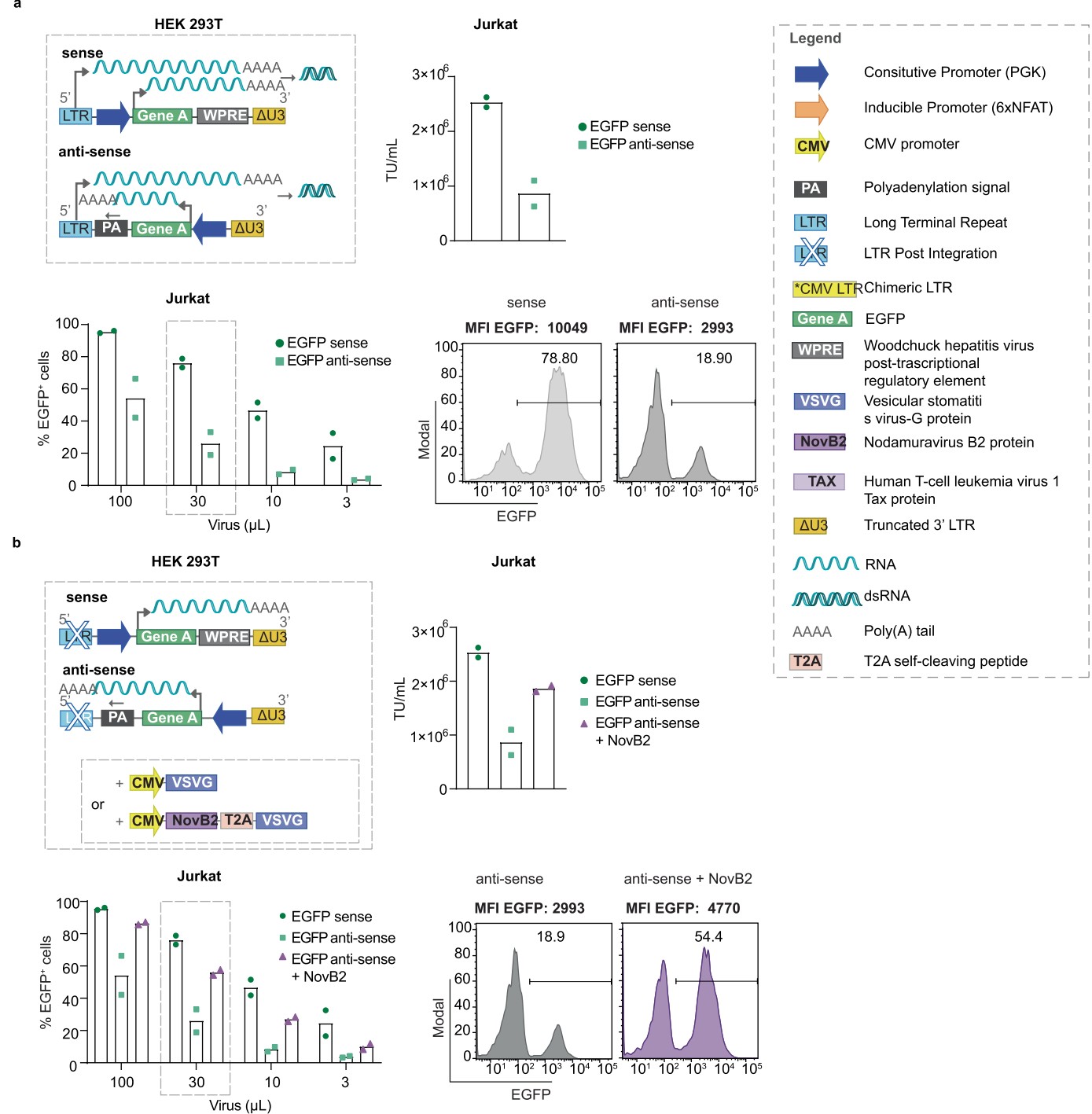

**Extended Data Fig. 3 | Antisense lentiviral transfer vector yields lower lentiviral titer than sense transfer vector and this can partially be restored by NovB2. (a) Top left**; schematic of sense and antisense constructs encoding EGFP only. **Top right**; titer measurement expressed as Transducing Units (TU) per ml, for two independent experiments. **Bottom left**; transduction of Jurkat cells with decreasing volumes of lentivirus vector supernatant to evaluate % EGFP expression by flow cytometric analysis on day 5. Bar graph represents the mean of two independent experiments. **Bottom right**; representative histograms of Jurkat cells transduced with 30 µl sense and antisense lentivirus vector supernatant. **(b) Top left**; schematic of sense and antisense orientation lentiviral

transfer vectors encoding EGFP post-integration in transduced cells. Antisense lentiviral vector was produced in the absence or presence of NovB2 (encoded on the envelope plasmid). **Top right**; titer measurement expressed as Transducing Units (TU) per ml for two independent experiments. **Bottom left**; transduction of Jurkat cells with decreasing volumes of lentivirus vector supernatant to evaluate % EGFP expression by flow cytometric analysis on day 5. Bar graph shows the mean of two independent experiments. **Bottom right**; representative histograms of Jurkat cells transduced with 30 µl anti-sense lentiviral vector supernatant produced in the absence or presence of NovB2.

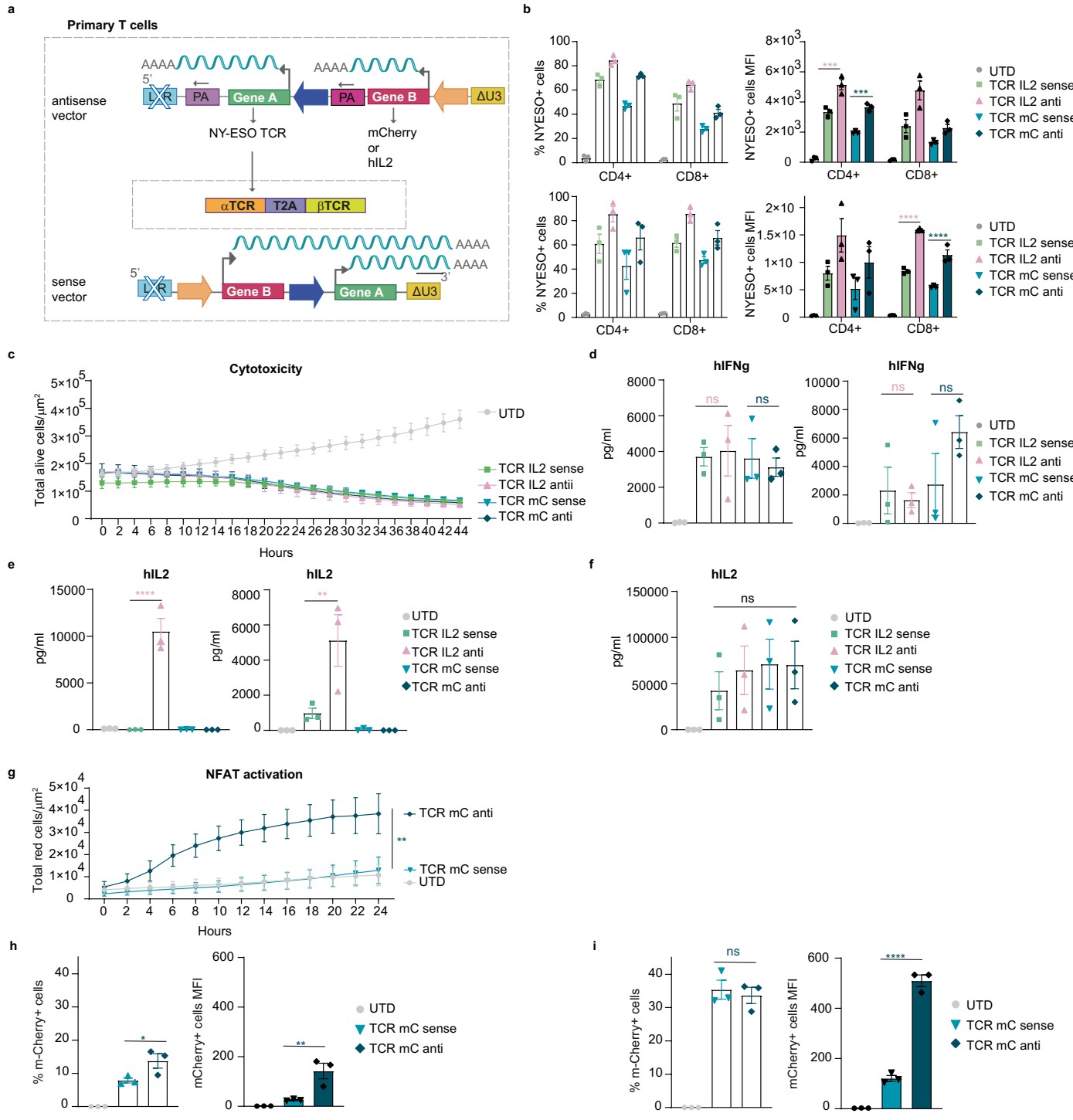

**Extended Data Fig. 4 | See next page for caption.**

**Extended Data Fig. 4 | Higher levels of inducible gene-cargo are produced by TCR-T cells transduced with the dual antisense versus sense lentiviral vector. (a)** Schematic of sense and antisense constructs encoding an HLA-A2 restricted NY-ESO-1$_{157-165}$ specific TCR (Gene A) 1 under the control of the PGK promoter and mCherry or h-IL-2 (Gene B) under the 6xNFAT promoter. **(b) Top and bottom left**; percentage TCR expression as measured by tetramer staining of primary human CD4$^+$ and CD8$^+$ T cells transduced with sense and antisense lentivirus vector supernatant produced in the presence of TNFα and NovB2. **Top and bottom right**; TCR expression levels (MFI values) for primary human CD4$^+$ and CD8$^+$ T cells transduced with sense and antisense lentivirus vector supernatant produced in the presence of TNFα and NovB2. Bar graph shows the mean + /- S.E.M. for $n$ = 6 human donors for two independent experiments ($n$ = 3 per experiment); ***$P$ < 0,001 upper panel TCR sense versus TCR IL2 antisense; ****$P$ < 0,001 bottom panel TCR IL2 sense versus TCR IL2 antisense) **(c)** Killing assay for TCR-T cells and UTD T cells against A2$^+$/NY$^+$ Saos-2 tumour cells labelled with nuclei green at ratio of 2:1 as measured by the IncuCyte instrument over time. Loss of total green area/μm$^2$ is proportional to killing activity. Shown are mean values + /- S.E.M. for T cells from $n$ = 3 human donors. **(d)** IFNγ quantification by ELISA assay of TCR- and UTD-T cells co-cultured with A2$^+$/NY$^+$ Saos-2 tumour cells at ratio of 2:1. Shown are mean values + /- S.E.M. for T cells from $n$ = 6 human donors for two independent experiments (left panel ns=0,9392 TCR IL2 sense versus TCR IL2 antisense, ns>0,9999 TCR mC sense versus TCR mC antisense; right panel ns=0,9959 TCR IL2 sense versus TCR IL2 antisense, ns=0,3562 TCR mC sense versus TCR mC antisense). **(e)** Human (h)

IL-2 quantification by ELISA assay of TCR- and UTD-T cells co-cultured with A2$^+$/NY$^+$ Saos-2 tumour cells at a ratio of 2:1. Shown are mean values + /- S.E.M. for T cells from $n$ = 6 human donors for two independent experiments (panel left ****$P$ < 0,0001 TCR IL2 sense versus TCR IL2 antisense; panel right **$P$ < 0,0023 TCR IL2 sense versus TCR IL2 antisense;). **(f)** hIL-2 quantification by ELISA assay of TCR- and UTD-T cells cultured overnight in the presence of PMA-Ionomycin (left panel ns=0,953 TCR IL2 sense versus TCR IL2 antisense, ns>0,9999 TCR mC sense versus TCR mC antisense). **(g)** Induced mCherry expression by TCR- and UTD-T cells against A2$^+$/NY$^+$ Saos-2 tumour cells at a ratio of 2:1 as measured by the IncuCyte instrument (total red area/μm$^2$) over time. Shown are mean values + /- S.E.M. for T cells from $n$ = 6 human donors (**$P$ = 0,0031 TCR mC sense versus TCR mC antisense at endpoint). **(h)** Flow cytometric analysis of mCherry expression for TCR- and UTD-T cells cells co-cultured with A2$^+$/NY$^+$ Saos-2 tumour cells at a ratio of 2:1. Shown are mean values + /- S.E.M. for T cells from $n$ = 3 human donors. **Left**; percentage of mCherry$^+$ cells (*$P$ = 0,0461 TCR mC sense versus TCR mC antisense). **Right**; mCherry expression levels (MFI) (**$P$ = 0,0092 TCR mC sense versus TCR mC antisense). **(i)** Flow cytometric analysis of mCherry expression for TCR- and UTD-T cells after overnight stimulation with PMA-Ionomycin. Shown are the mean values + /- S.E.M. for T cells from $n$ = 3 human donors **Left**; percentage of mCherry$^+$ cells (ns=0,8478 TCR mC sense versus TCR mC antisense). **Right**; mCherry expression levels (relative MFI) (****$P$ < 0,0001 TCR mC sense versus TCR mC antisense). Two-way (panel c and g) and One-way Anova (panels b,d,e,f,h and i) tests were used to determine statistical significance.

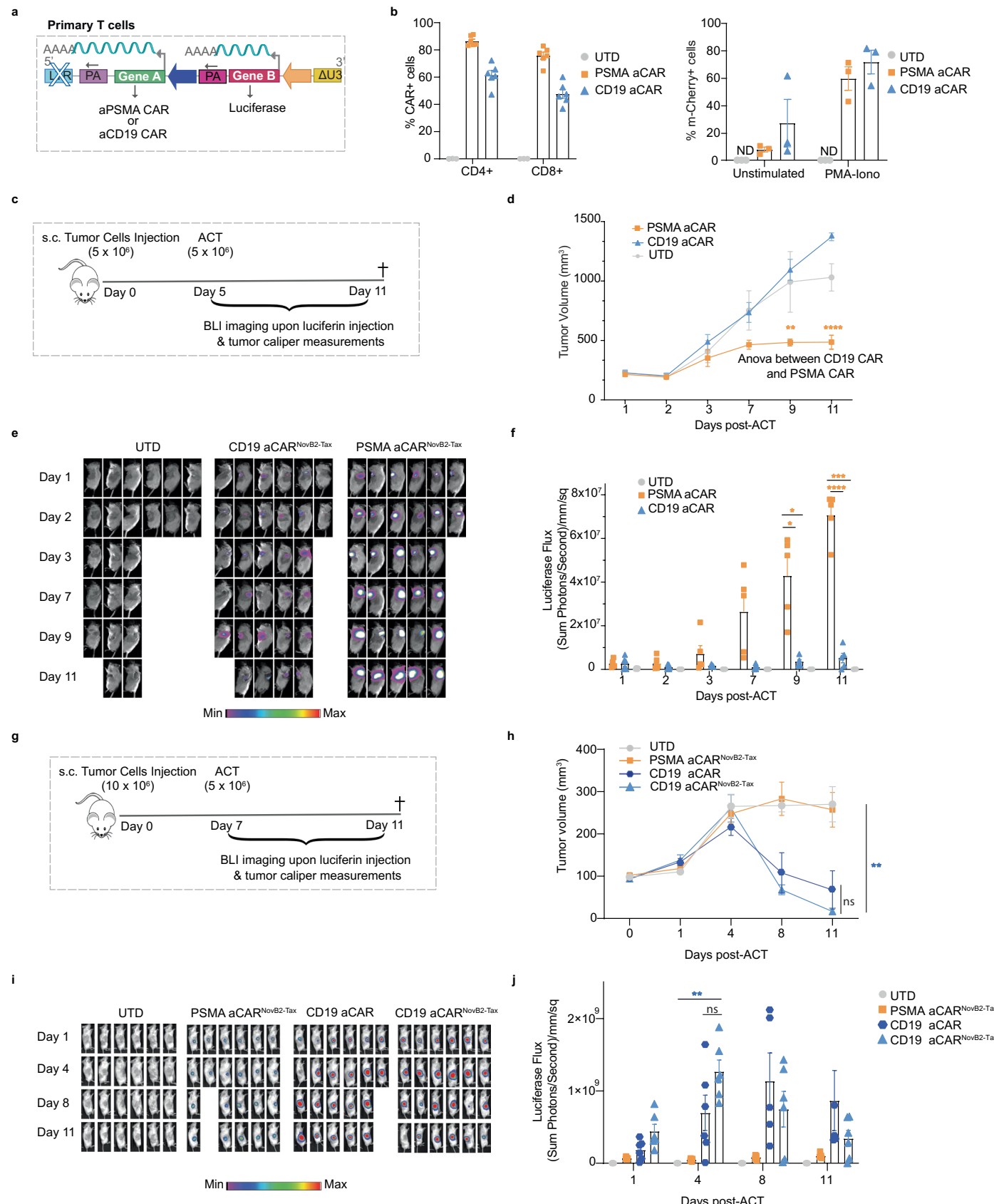

**Extended Data Fig. 5 | See next page for caption.**

**Extended Data Fig. 5 | T cells transduced with antisense lentiviral vector encoding a CAR and inducible gene-cargo demonstrate specific in vitro and in vivo function and are not impacted by the use of NovB2 and Tax during virus production. (a)** Schematic of sense and antisense lentiviral vector encoding the anti-PSMA and anti-CD19 CARs under the PGK promoter and firefly luciferase under 6xNFAT. **(b) Left**; Transduction efficiency of CD4$^+$ and CD8$^+$ primary T cells as measured by cell-surface CAR expression. Bar graphs show the mean +/- S.E.M. of the percentage of CAR$^+$ T cells. Data are for T cells from n = 6 healthy donors and symbols on the graphs represent individual donors. **Right;** mCherry expression at 12 h post PMA-Ionomycin stimulation by equivalently transduced T cells as measured by flow cytometric analysis. With the T cells normalized to approximately 40% cell-surface CAR-expression the graph indicates that all transduced T cells express mCherry upon activation by PMA-Ionomycin. Shown are mean values +/- S.E.M. Symbols indicate individual donors (n = 3) **(c)** Schematic of CAR-T cell transfer study in PSMA$^+$ PC3-PIP tumour bearing mice. **(d)** Caliper tumour volume measurements over days. Values are the mean +/- S.E.M. for n = 5 mice per group. Statistical significance was determined by Two-way ANOVA. (Day9 **$P$ = 0,034 aPSMA versus aCD19;

Day11 ****$P$ < 0,0001 aPSMA versus aCD19) **(e)** Representative images of luciferase activity of the transferred T cells over days upon luciferin injection in mice. **(f)** Bar graph shows the mean value of luciferase flux for all experimental groups. Data are represented as the mean +/- S.E.M. and for n = 5 mice per group. Statistical significance was assessed using a Two-Way ANOVA and Post-hoc Tukey test. (Day9 *$P$ = 0,0201 aPSMA versus aCD19, *$P$ = 0,0156 aPSMA versus UTD; Day11****$P$ < 0,0001 aPSMA versus aCD19, ***$P$ = 0,0003 aPSMA versus UTD). **(g)** Schematic of CAR-T cell transfer study in CD19$^+$ Bjab tumour bearing mice. **(h)** Caliper tumour volume measurements over days. Values are the mean +/- S.E.M. for n = 6 mice per group. Statistical significance was determined by Two-way ANOVA (Day11 ns=0,726 aCD19 versus aCD19 Tax-NovB2, **$P$ = 0,0051 aCD19 Tax-NovB2 versus UTD. **(i)** Representative images of luciferase activity of the transferred T cells over days upon luciferin injection in mice. **(j)** Bar graph shows the mean value of luciferase flux for all the experimental groups. Data are represented as the mean +/- S.E.M. and for n = 6 mice per group. Statistical significance was assessed using Two-Way ANOVA and Post-hoc Tukey test. (Day11 ns=0,1589 aCD19 versus aCD19 Tax-NovB2, **p = 0,0019 aCD19 Tax-NovB2 versus UTD. (ns = non-significant, **$P$ < 0.01 and ****$P$ < 0.0001).

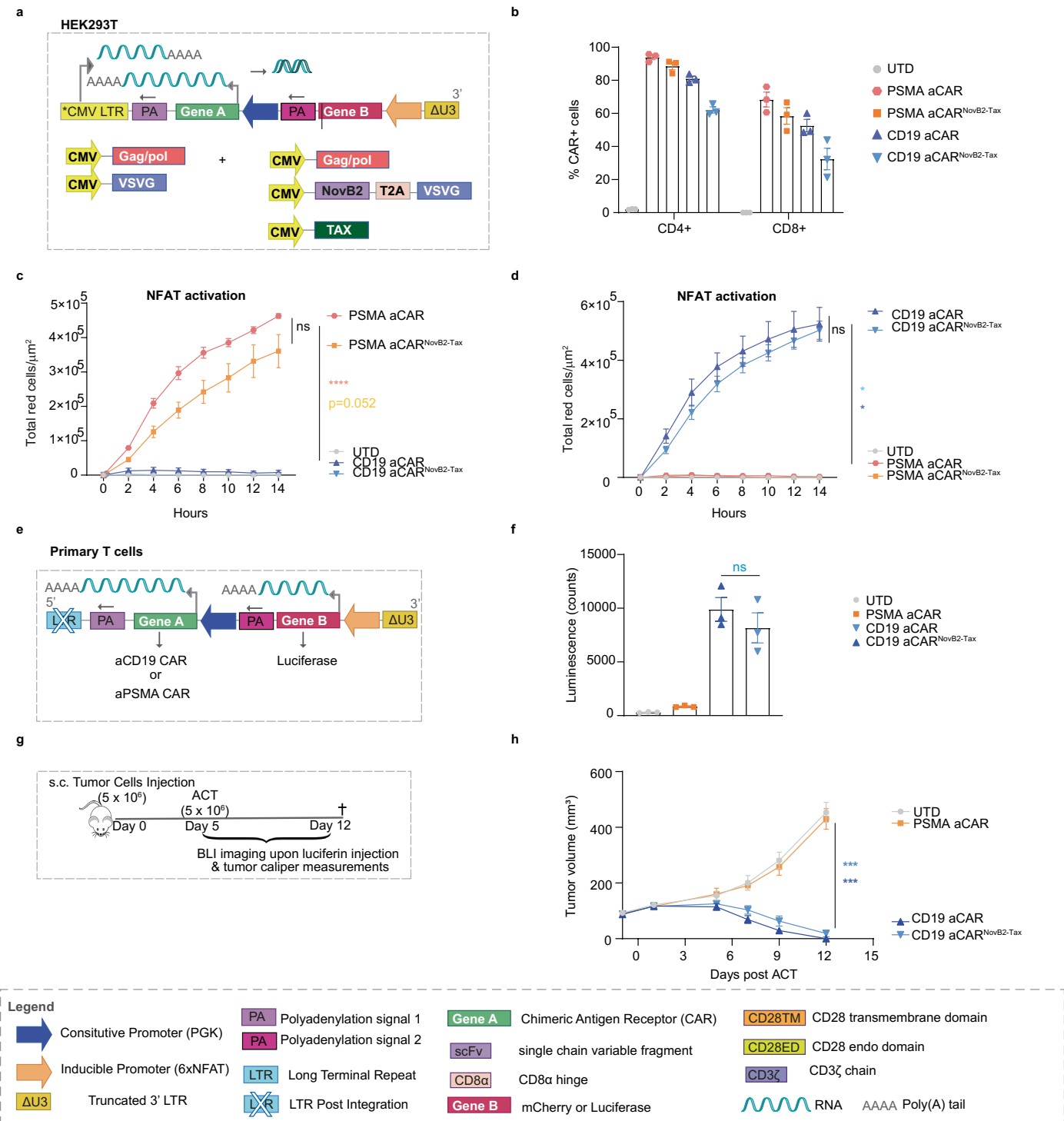

**Extended Data Fig. 6 | The production of antisense lentiviral vector in the presence of NovB2 and Tax does not impact the activity levels of transduced T cells. (a)** Schematic of antisense lentiviral vector encoding the anti-PSMA (Gene A) and anti-CD19 (Gene B) CARs under the PGK promoter along with mCherry under 6xNFAT. **(b)** Transduction efficiency of CD4+ and CD8+ primary T cells using lentiviral supernatant produced in absence or presence of both NovB2 and Tax. Bar graphs show the mean +/- S.E.M. of percentage of CAR+ T cells. Data shown are for T cells from n = 3 human donors and symbols represent individual donors. **(c)** Evaluation of mCherry expression (total red area/μm²) by activated anti-PSMA and **(d)** Evaluation of mCherry expression (total red area/μm²) by activated anti CD19 CAR-T cells upon co-culture with PSMA + PC3-PIP tumour cells (panel c ns=0,4921 aPSMA versus aPSMA Tax-NovB2, ****P < 0,0001 aPSMA versus UTD, P = 0,0525 aPSMA Tax-NovB2 versus UTD); (panel d ns=0,9671 aCD19 versus aCd19 Tax-NovB2, *P < 0,0361 aCD19 versus UTD,**P = 0,0121 aCD19 Tax-

NovB2 versus UTD). Values for the IncuCyte assay are the mean ± S.E.M. for T cells from n = 3 human donors. Statistical significance was assessed using a Two-Way ANOVA and Post-hoc Tukey test. **(e)** Schematic of antisense lentiviral vectors encoding the anti-PSMA or anti-CD19 CARs (Gene A) and luciferase as gene cargo (Gene B). The CARs are expressed under the PGK promoter and luciferase under 6xNFAT. **(f)** Induction of luciferase in anti-CD19 CAR-T cells upon 24 h co-culture with PC3-CD19+ tumour cells. Bar graph represents mean +/- S.E.M. of luminescence (counts) measured by HIDEX. Data are for T cells from n = 3 human donors. (ns=0,5563 aCD19 versus aCD19 Tax-NovB2). **(g)** Schematic of CAR-T cell transfer study in PC3-CD19 tumour bearing mice. **(h)** Caliper tumour volume measurements over days. Values are the mean +/- S.E.M. for n = 6 mice per group. Statistical significance was determined by Two-way ANOVA (Day12 ns=0,46 aCD19 versus aCd19 Tax-NovB2, ***P < 0,001 aCD19 versus UTD,***P < 0,001 aCD19 Tax-NovB2 versus UTD.

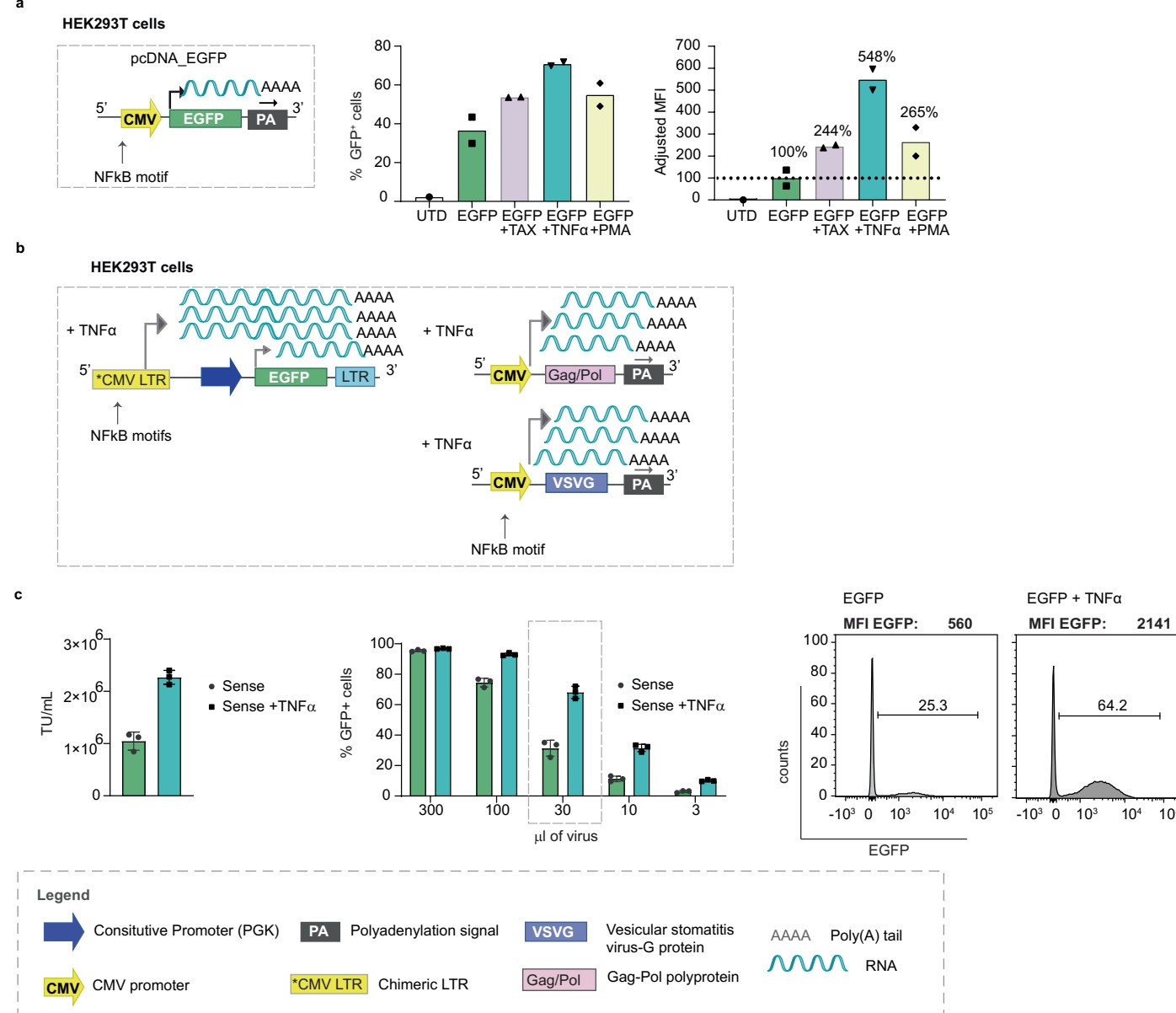

**Extended Data Fig. 7 | TNFα can be used instead of Tax to augment transcription from vectors comprising a CMV promoter. (a) Left**; schematic of pcDNA plasmid encoding EGFP under a CMV promoter in the sense orientation. **Middle**; bar graph representing % EGFP expressing HEK293T cells 48 hours after transfection with suboptimal levels of plasmid in the presence or not of co-transfected plasmid encoding Tax, soluble TNFα, or PMA. **Right**; bar graph shows relative mean fluorescence intensity (MFI) of EGFP under the different experimental conditions (EGFP encoding plasmid alone is set to 100%). **(b)**

Schematic of sense lentiviral vector encoding EGFP and produced in absence or presence of TNFα **(c) Left**; titer measurement expressed as Transducing Units (TU) per ml for three independent experiments. **Middle**; transduction of Jurkat cells with decreasing volumes of lentivirus vector supernatant to evaluate percentage EGFP expression by flow cytometric analysis on day 5. Bar graph represents the mean of three independent experiments. **Right**; representative histograms of Jurkat cells transduced with 30 μl sense and antisense lentivirus vector supernatant.

| | |
|---|---|

# Reporting Summary

## Statistics

For all statistical analyses, confirm that the following items are present in the figure legend, table legend, main text, or Methods section.

| n/a | Confirmed | |
|---|---|---|
| ☐ | ☒ | The exact sample size (*n*) for each experimental group/condition, given as a discrete number and unit of measurement |
| ☐ | ☒ | A statement on whether measurements were taken from distinct samples or whether the same sample was measured repeatedly |
| ☐ | ☒ | The statistical test(s) used AND whether they are one- or two-sided<br>*Only common tests should be described solely by name; describe more complex techniques in the Methods section.* |
| ☒ | ☐ | A description of all covariates tested |
| ☐ | ☒ | A description of any assumptions or corrections, such as tests of normality and adjustment for multiple comparisons |
| ☐ | ☒ | A full description of the statistical parameters including central tendency (e.g. means) or other basic estimates (e.g. regression coefficient) AND variation (e.g. standard deviation) or associated estimates of uncertainty (e.g. confidence intervals) |
| ☐ | ☒ | For null hypothesis testing, the test statistic (e.g. *F*, *t*, *r*) with confidence intervals, effect sizes, degrees of freedom and *P* value noted<br>*Give P values as exact values whenever suitable.* |
| ☒ | ☐ | For Bayesian analysis, information on the choice of priors and Markov chain Monte Carlo settings |
| ☒ | ☐ | For hierarchical and complex designs, identification of the appropriate level for tests and full reporting of outcomes |
| ☒ | ☐ | Estimates of effect sizes (e.g. Cohen's *d*, Pearson's *r*), indicating how they were calculated |

*Our web collection on statistics for biologists contains articles on many of the points above.*

## Software and code

Policy information about availability of computer code

| | |
|---|---|
| Data collection | Incucyte Instrument, BD LSR II FACS, BD LSR SORP FACS, In-Vivo Xtreme system (Bruker Corp.), Western Blot Imager (Fusion, Vilber Lourmat). HIDEX. |
| Data analysis | IncuCyte Zoom 2016A Data analysis (Essen Bioscience), FACS DIVA Software, Microsoft Excel 2019, GraphPad Prism v8, FlowJo X, Molecular Imaging (MI) software (MI, Bruker Corp.), ImageJ software (pixel intensity of the bands). Hidex software for bioluminescence. |

For manuscripts utilizing custom algorithms or software that are central to the research but not yet described in published literature, software must be made available to editors and reviewers. We strongly encourage code deposition in a community repository (e.g. GitHub). See the Nature Portfolio guidelines for submitting code & software for further information.

## Data

Policy information about availability of data

All manuscripts must include a data availability statement. This statement should provide the following information, where applicable:
- Accession codes, unique identifiers, or web links for publicly available datasets
- A description of any restrictions on data availability
- For clinical datasets or third party data, please ensure that the statement adheres to our policy

> The main data supporting the findings of this study are available within the article and its Supplementary Information. Source data for the figures are provided with this paper. All raw data generated during the study are available from the corresponding authors on request.

# Field-specific reporting

Please select the one below that is the best fit for your research. If you are not sure, read the appropriate sections before making your selection.

☒ Life sciences ☐ Behavioural & social sciences ☐ Ecological, evolutionary & environmental sciences

For a reference copy of the document with all sections, see nature.com/documents/nr-reporting-summary-flat.pdf

# Life sciences study design

All studies must disclose on these points even when the disclosure is negative.

| | |
|---|---|
| Sample size | For the in vitro studies, the number of healthy donors and/or technical replicates were chosen according to the complexity of the assay and for the expected biological variability. For the in vivo studies, a maximum of 5 million CAR T cells per mouse were needed. We achieved a sample size of 5 animal per treatment group, which proved to be sufficient to reproducibly observe statistically significant differences. |
| Data exclusions | For the in vivo studies, upon calipering, outlier mice with extreme burdens (either too high or too low compared to the average) were excluded from the experiment before CAR-T-cell transfer. No mice were excluded afterwards at any point. |
| Replication | All attempts at replication were successful. |
| Randomization | Tumor burden was evaluated by calipering the same day of CAR T cell transfer. Outlier mice with extreme burdens (either too high or too low compared to the average) were excluded from the experiment before CAR T cell transfer. No mice were excluded afterwards at any point. Following tumor burden measure, mice were assigned into treatment groups such that each group had the same overall average tumor volume. Buffy coats and apheresis filters were obtained from anonymous donors. |
| Blinding | An independent investigator verified caliper measurements in a blinded fashion. The analysis of data (the plotting of pre-recorded tumour volumes at end of study) was performed in a non-blinded manner. |

# Reporting for specific materials, systems and methods

We require information from authors about some types of materials, experimental systems and methods used in many studies. Here, indicate whether each material, system or method listed is relevant to your study. If you are not sure if a list item applies to your research, read the appropriate section before selecting a response.

## Materials & experimental systems

| n/a | Involved in the study |
|---|---|
| ☐ | ☒ Antibodies |
| ☐ | ☒ Eukaryotic cell lines |
| ☒ | ☐ Palaeontology and archaeology |
| ☐ | ☒ Animals and other organisms |
| ☐ | ☒ Human research participants |
| ☒ | ☐ Clinical data |
| ☒ | ☐ Dual use research of concern |

## Methods

| n/a | Involved in the study |
|---|---|
| ☒ | ☐ ChIP-seq |
| ☐ | ☒ Flow cytometry |
| ☒ | ☐ MRI-based neuroimaging |

# Antibodies

| | |
|---|---|
| Antibodies used | Antibodies were titrated for optimal staining. Aqua live Dye BV510 (cat L34966, lot no 1899019, Invitrogen ) and near-IR fluorescent reactive dye (APC Cy-7) (cat L34976A, Invitrogen, lot no 2379385) were used to assess viability. AlexaFluo 647-conjugated anti-mouse F(ab)' (cat 115-606-072, lot no 143040, Jackson Immuno Research) was used fror CAR detection. PE-labeled anti-human PSMA (clone LNI-17, cat 342503, lot no B211499, Biolegend) and PE Isotype mouse-anti-IgG1, k chain (clone MOPC-21, cat 400114, lot no B307873, Biolegend) were used to evaluated PSMA expression on cell line. PE-labeled HLA-A2/NY-ESO-1(157-165) tetramer (batch number 011220, PTCF at UNIL) was used to evaluate TCR expression. PE-labeld anti human NGFR (clone ME20.4, cat 345105, lot no B262596, Biolegend) was used to evaluate transduction efficiency. Anti-human HPK1 (rabbit monoclonal clone EP6430Y, cat ab33910, Abcam) was used to detect MAP4K1/HPK1 levels with western blot analysis. Anti-human-beta Actin (cat sc-47778, Santa Cruz) was used to detect beta-Actin as control for wester blot. PE-labled anti-human PAN TCR (cat B49177, lot no 200029, Beckman Coulter) was used to detect TCR expression on human T cells. |
| Validation | Antibody-concentration validation was empirically determined in the lab. |

# Eukaryotic cell lines

Policy information about <u>cell lines</u>

| | |
|---|---|
| Cell line source(s) | 293T, Jurkat, Saos, A375, Me275, Na8 and Bajb from ATCC. PC3-PIP from Dr. Rosato, University of Padova. Jurkat 6x-NFAT mCherry were engineered in the lab. PC3 and PC3-PIP CD19+ were engineered in the lab, provided by Prof. Jannick Muller. |
| Authentication | COA provided with cell line by ATCC. Properties pertinent to experiment (such as PSMA expression) were confirmed by flow cytometry. |
| Mycoplasma contamination | All cell lines were routinely tested for mycoplasma contamination, and found to be negative. |
| Commonly misidentified lines (See <u>ICLAC</u> register) | No commonly misidentified cell lines were used. |

# Animals and other organisms

Policy information about <u>studies involving animals</u>; <u>ARRIVE guidelines</u> recommended for reporting animal research

| | |
|---|---|
| Laboratory animals | NSG male mice, 8–12 weeks old, were bred and housed in a SOPF animal facility. |
| Wild animals | The study did not involve wild animals. |
| Field-collected samples | The study did not involve samples collected from the field. |
| Ethics oversight | All in vivo experiments were conducted in accordance with and approval from the Service of Consumer and Veterinary Affairs (SCAV) of the Canton of Vaud (Switzerland), license VD3414. |

Note that full information on the approval of the study protocol must also be provided in the manuscript.

# Human research participants

Policy information about <u>studies involving human research participants</u>

| | |
|---|---|
| Population characteristics | Buffy coats and apheresis filters from anonymous healthy donors were collected with informed consent of the donors, and genetically engineered with Ethics Approval from the Canton of Vaud. |
| Recruitment | N/A |
| Ethics oversight | Ethics approval from the Canton of Vaud to the laboratory of Prof. George Coukos allowed gene engineering of primary human T cells. |

Note that full information on the approval of the study protocol must also be provided in the manuscript.

# Flow Cytometry

## Plots

Confirm that:

☒ The axis labels state the marker and fluorochrome used (e.g. CD4-FITC).

☒ The axis scales are clearly visible. Include numbers along axes only for bottom left plot of group (a 'group' is an analysis of identical markers).

☒ All plots are contour plots with outliers or pseudocolor plots.

☒ A numerical value for number of cells or percentage (with statistics) is provided.

## Methodology

| | |
|---|---|
| Sample preparation | Primary human T cells were isolated from the peripheral blood mononuclear cells (PBMCs) of healthy donors (HDs; prepared as buffycoats or apheresis filters). All blood samples were collected with informed consent of the HDs, and genetically-engineered with Ethics Approval from the Canton of Vaud to the laboratory of Prof. George Coukos. Total PBMCs were obtained via Lymphoprep (Axonlab) separation solution, using a standard protocol of centrifugation. CD4+ and CD8+ T cells were isolated using a magnetic bead-based negative selection kit following the manufacturer's recommendations (easySEP, Stem Cell technology). Purified CD4+ and CD8+ T cells were cultured at a 1:1 ratio in RPMI-1640 with Glutamax, supplemented with 10% heat-inactivated FBS, 100 U/mL penicillin, 100 μg/mL streptomycin sulfate, and stimulated with anti-CD3 and anti-CD28 monoclonal antibody (mAb)-coated-beads (Lifetechnologies) in a ratio of 1:2, T cells: beads. For staining preparation, cells were washed once and resuspended in FACS buffer containing LIVE/DEAD dye and the antibody cocktail. Cells were incubated at 4 degrees for 30 minutes and washed twice before acquisition. Cells were not fixed prior to acquisition. |
| Instrument | LSR II, BD |

| Software | Collection: FACS DIVA<br>Analysis: FlowJo X |
|---|---|
| Cell population abundance | T-cell purification from PBMCs by magnetic beads was validated by flow for CD4+/CD8+ cells. T-cell purity was >99%. |
| Gating strategy | The starting cell population was gated on a linear SSC-A/FSC-A plot. Single cells were discriminated on a linear FSC-H/FSC-A plot. Live cells were determined by exclusion from positive live/dead-stained cells. Positive/negative populations were determined with negative controls. |

☐ Tick this box to confirm that a figure exemplifying the gating strategy is provided in the Supplementary Information.

