## [Peer Review File · Nature Biomedical Engineering]

A lentiviral vector for the production of T cells with an inducible transgene and a constitutively expressed tumour-targeting receptor

Corresponding author: Melita Irving

Editorial note

This document includes relevant written communications between the manuscript's corresponding author and the editor and reviewers of the manuscript during peer review. It includes decision letters relaying any editorial points and peer-review reports, and the authors' replies to these (under 'Rebuttal' headings). The editorial decisions are signed by the manuscript's handling editor, yet the editorial team and ultimately the journal's Chief Editor share responsibility for all decisions.

Any relevant documents attached to the decision letters are referred to as **Appendix #**, and can be found appended to this document. Any information deemed confidential has been redacted or removed. Earlier versions of the manuscript are not published, yet the originally submitted version may be available as a preprint. Because of editorial edits and changes during peer review, the published title of the paper and the title mentioned in below correspondence may differ.

Correspondence

Wed 11 May 2022

Decision on Article nBME-22-0571

Dear Dr Irving,

Thank you again for submitting to *Nature Biomedical Engineering* your manuscript, "Optimised inverted lentiviral transfer vector comprising two independent promoters enabling on command gene-cargo delivery by tumor redirected T cells". The manuscript has been seen by three experts, whose reports you will find at the end of this message. You will see that the reviewers appreciate the work, and that they raise a number of technical criticisms that we hope you will be able to address. In particular, we would expect that a revised version of the manuscript provides:

- * Direct measurements of the claimed higher titres of the optimized inverted lentiviral vector.
- * Discussion, supported with evidence, of the differences and any advantages of your vector and existing vectors that also deliver a constitutively expressed CAR and an inducible gene-expression cassette that is activated on recognition of the CAR's target antigen (such as the work in your ref. 34).
- * Thorough methodological and statistical reporting, as per the relevant comments of all reviewers.

When you are ready to resubmit your manuscript, please upload the revised files, a point-by-point rebuttal to the comments from all reviewers, the reporting summary, and a cover letter that explains the main improvements included in the revision and responds to any points highlighted in this decision.

Please follow the following recommendations:

- * Clearly highlight any amendments to the text and figures to help the reviewers and editors find and understand the changes (yet keep in mind that excessive marking can hinder readability).- * If you and your co-authors disagree with a criticism, provide the arguments to the reviewer (optionally, indicate the relevant points in the cover letter).
- * If a criticism or suggestion is not addressed, please indicate so in the rebuttal to the reviewer comments and explain the reason(s).
- * Consider including responses to any criticisms raised by more than one reviewer at the beginning of the rebuttal, in a section addressed to all reviewers.
- * The rebuttal should include the reviewer comments in point-by-point format (please note that we provide all reviewers will the reports as they appear at the end of this message).
- * Provide the rebuttal to the reviewer comments and the cover letter as separate files.

We hope that you will be able to resubmit the manuscript within 16 weeks from the receipt of this message. If this is the case, you will be protected against potential scooping. Otherwise, we will be happy to consider a revised manuscript as long as the significance of the work is not compromised by work published elsewhere or accepted for publication at *Nature Biomedical Engineering*.

We hope that you will find the referee reports helpful when revising the work, which we look forward to receive. Please do not hesitate to contact me should you have any questions.

Best wishes,

Pep

Pep Pàmies
Chief Editor, Nature Biomedical Engineering

Reviewer #1 (Report for the authors (Required)):

Reichenbach and colleagues designed an inverted lentiviral transfer vector, which combined with renewed strategies to enhance virus production significantly improves the efficiency of T cell engineering to provide these cells with gene cargo. These findings represent highly relevant and timely progress to the field of adoptive T cell therapy enabling the making of more effective cellular products to treat solid tumors.

Recommendations:

(1) The introduction is biased towards CAR T cells. Given the relevance of co-engineering for both CAR and TCR T cells alike, as well as data presented in Figure 5, I recommend to include TCR T cells from the start along with relevant references regarding co-engineering of the latter cells.

(2) Conclusions regarding anti-sense being the preferred design for transfer vectors are fair, yet authors should acknowledge that the head-to-head comparison of the 3 designs is challenged by the non-ability to use PA sites in the sense design. In addition, the use of different PA sites (BGH vs synthetic) and use or not of a transcription pausing site further complicates direct comparisons. Authors should discuss this.

(3) With respect to the NOVB2 and TAX-based strategies to overcome low LV titers when using anti-sense vectors, it would provide surplus value when authors present measurements of actual titers rather than using eGFP as a surrogate marker. Specifically for TAX, and again making direct comparisons difficult, authors should justify why they switch to a CMV promoter and leave out Gene B in their tests.

(4) When following up the NOVB2 and TAX strategies with in vitro and in vivo testing of CAR T cells, authors need to include a control using single CAR vectors. Reason being: when using 'gene additives' in the HEK production phase, one formally cannot exclude that such additives are passed onto T cells potentially

affecting expression and function of the CAR. In addition, the inclusion of CD19 CAR T cells vs CD19-positive targets (and using the PSMA CAR as a negative control) would provide additional robustness to findings.

(5) The inclusion and testing of siRNA reads as a 'turning point'. I would rather first see an optimal protocol for transgenic cargos. Authors should consider two main parts, first finalizing the protocol for transgenes, second go into knock-down applications. Along these lines, it would be better to substantiate part 1 with 'real' gene cargo (i.e., IL12). Leave out knock-down of TRAC as this is now combined w/ eGFP rather than a TCR (consequently not assessing the effect on TCR mis-pairing), and focus on just 1 knock-down example, being Hpk-1. Regarding the 2nd part (as has been done for the first part), include in vitro function of T cells, not trivial in the context of Hpk1 deficiency and its effect on TCR signaling.

Minor points:

- Manuscript is well written, although here and there at an educational level. Please make it less tutorial (many explanations between brackets are no necessity).
- Discussion: avoid repetition of introduction. Rather discuss limitations of proposed methods, and what experiments would be needed to implement these methods in the clinical setting. Also, authors should discuss why CD4 T cells appear more prone for these transfer vector-mediated transductions when compared to CD8 T cells?
- In line with the above recommendation to split Manuscript into 2 parts, Figure 5 could go supplementary.
- When measuring the 2nd gene product upon activation w/ CAR T cells (Figure 4c), it would be better to replace PMA/I by an anti-CAR stimulus.
- Figures: F4: include the NovB2 and TAX methodology in cartoons; F4c: CAR transductions are normalized to 40% (in text it states 50%); F6a has little additional value over F5b; F6c, cartoon, CAR should be replaced by eGFP; and SF1, axes w/ dot plots are not annotated.

Reviewer #2 (Report for the authors (Required)):

This is an interesting manuscript in which the authors explore the optimal configuration of lentiviral vectors to express two transgenes (constitutive x2 or constitutive and inducible post activation), or a siRNA and a transgene. The authors use a very methodological approach and the results are convincing although the 'n' of most experiments seems very low. However, my biggest concern centers around novelty and innovation.

Major concerns

- 1) No new promoters or conceptually new vector designs are being explored or being developed. For example, one of the developed lentiviral vectors enables the expression of siRNAs and a transgenes. Retroviral vectors with the same capabilities were developed more than a decade ago (PMID: 23250361, PMID: 21673345).
- 2) It seems that many experiments were only performed once or only in technical replicates (e.g., Figs 1-
- 3). It is unclear how many different human donors were used to conduct the experiments.

Minor concerns

- 1) While titration experiments with lentiviral vectors are performed, no formal titers are calculated. This is critical to assess if the developed lentiviral constructs are suitable for clinical translation.

Reviewer #3 (Report for the authors (Required)):

In their manuscript, Reichenbach and colleagues describe the design of an inverted lentiviral vector that contains one promoter to constitutively express one gene (e.g. a chimeric antigen receptor (CAR) to direct T cell activity against tumor cells expressing target antigens) and a second promoter that acts in an inducible manner by responding to signals that derive from activation of the constitutively expressed CAR construct. This work is reminiscent of earlier descriptions of armored CARs and TRUCK concepts that also used NFAT response elements coupled with the minimal IL2 promoter (as the authors also correctly reference).

The authors show some of the optimization steps they used to decrease non-specific activation of the

inducible gene cassette component of this newly designed vector system. In the abstract, the authors promise reduced lentiviral production costs as this is a single vector system that can be used to generate higher titers. As also referenced by the authors (e.g. refs 34 and 35), other single vector systems for such purposes were published earlier. To address the claim that higher lentiviral vector titers can be achieved with the vector system described in this manuscript. The authors do show that co-delivery of NovB2 and Tax (later substituted for TNF α) proteins can be used to increase transduction efficiencies of the lentiviral vectors described here, however, this effect is difficult to assess in the manner that the data is presented. In several instances, the authors provide volumes of lentiviral vector supernatants that were applied for the transduction experiments. The

authors should provide some calculation of vector titers for comparison to other established systems. This is especially important as the quality / infectivity of different lentiviral vector productions can vary due to several factors.

In general, such approaches to develop synthetic vector systems for improved modification of immune cells is of high interest and could have impact on the fields of immunotherapy, virology and retroviral vector development and could even become clinically relevant if shown to be robust. Indeed, the authors present data showing transduction of primary human T cells with the inverted lentiviral vector that was designed to express CARs and that the CAR T cells were active against cancer cells in vitro and in vivo. While these findings might not be surprising as CAR constructs known to function were used, it does provide proof-of-principle that the current vector system can be successfully used to generate functional CAR T cells. In addition to the points mentioned already, please find some additional comments that should be addressed:

Major points to be addressed:

1. In figure 1, representative flow cytometric analyses are shown. It would be more informative and appropriate to include graphs that show the results for all experiments accomplished (i.e. more than the one experiment shown in panels a, b and c). This would provide the reader a better chance to evaluate the presented data. Also, in some figure legends, the authors write “representative” without providing further information – these data are representative of how many experiments?
2. As mentioned above, the authors describe transduction with volumes (μ l) of lentiviral vector supernatants. These should rather be shown as multiplicities of infection to apply similar amounts of infectious particles and/or quantification of lentiviral capsid or RNA. Otherwise, it is difficult to judge and compare the effectiveness of the lentiviral vector preparations.
3. What is the authors’ rationale for using the 30 μ l lentiviral vector supernatant amounts for comparison to claim improved transduction of Jurkat cells with the anti-sense vector in combination with NovB2 and Tax? This seems arbitrary as little or no improvement to transduction levels are seen with the combination when higher volumes of the lentiviral vector supernatants are used. The authors could add a sentence in the appropriate section of the results to make this more clear.
4. In figure 4, a comparison with the sense lentiviral vector with regard to lentiviral vector supernatant titers of the CAR vectors seems to be missing. In figure 4c, how do the authors explain the strong mCherry background expression in the unstimulated CAR T cells?
5. In the description of the data shown in figure 5, the authors write “important increase in titer”, but do not actually show any titers for the lentiviral vectors. These should be calculated and shown.

Minor points to be addressed:

1. In many instances throughout the manuscript, the authors write “lentivirus mediated”. I think this would be more correct as “lentivirus vector mediated”, as a vector derived from the lentivirus is used.
2. Sometimes the authors write “ml” in the figure legends describing the amount of lentiviral vector supernatant used. This is presumable μ l.
3. In supplemental figure 1, the labels for the flow cytometry plot axes are missing (i.e. eGFP and mCherry).
4. The manuscript reads mostly ok, but does need some editing for punctuation and language use (e.g. “ameliorate” is often used incorrectly).

Thu 24 Nov 2022

Decision on Article nBME-22-0571A

Dear Dr Irving,

Thank you for your revised manuscript, "Optimised inverted lentiviral transfer vector comprising two independent promoters enabling on command gene-cargo delivery by tumor redirected T cells", which has been seen by the original reviewers. In their reports, which you will find at the end of this message, you will see that the reviewers acknowledge the improvements to the work and that Reviewer #3 raises a few relatively minor technical criticisms that I am sure you will be able to address. In particular, please clearly state the limitations of the approach and, as the reviewer suggests, include the earlier data in Fig. 4c as Supplementary Information.

As before, when you are ready to resubmit your manuscript, please upload the revised files and a point-by-point response to the comments from Reviewer #3.

We look forward to receive a further revised version of the work. Please do not hesitate to contact me should you have any questions.

Best wishes,

Pep

Pep Pàmies
Chief Editor, Nature Biomedical Engineering

Reviewer #1 (Report for the authors (Required)):

Reichenbach and colleagues have profoundly and accurately addressed and/or rebutted all of my concerns that were raised in response to their original contribution. The revised manuscript shows improved quality and quantity of data, as well as a better overall structure, flow of reading and discussion of findings.

Reviewer #2 (Report for the authors (Required)):

I would like to thank the authors for submitting a substantially revised manuscript that addresses all my concerns. I would like to congratulate the authors to their important study.

Reviewer #3 (Report for the authors (Required)):

In the revised manuscript, Reichenbach and colleagues have addressed most of the concerns raised during the first round of review by adding additional data and further editing the text of the manuscript. In short, the manuscript is greatly improved, but there remain some issues to address.

Major technical point to address: The authors addressed the question about the strong mCherry background expression in unstimulated T cells shown in the original figure 4c by replacing this data with other experiments, but I think this might have been important data as it is always best to know potential drawbacks of an approach or further areas of possible improvement (such as to overcome leakiness of an inducible promoter). In the revised figure 4 panels g and h, non-stimulated controls to quantify the background mCherry reporter expression from the CAR-T cells seems to be missing.

Minor (but still important points to address): Although the authors now use "lentivirus vector" in some

instances in the modified manuscript, they do still quite often use the term “lentivirus” throughout the manuscript, which, is incorrect as the system described in this work is a lentivirus vector system. Thus, this still needs to be corrected in many instances.

Thu 12 Jan 2023

Decision on Article nBME-22-0571B

Dear Dr Irving,

Thank you for your patience in waiting for the guidelines for the final submission of your manuscript, "Inverted lentiviral transfer vector comprising independent promoters for on-command gene-cargo delivery by tumor redirected T cells" to *Nature Biomedical Engineering*. Please carefully follow the instructions provided in the attached file.

For primary research originally submitted after December 1, 2019, we encourage authors to take up transparent peer review. If you are eligible and opt in to transparent peer review, we will publish, as a single supplementary file, all the reviewer comments for all the versions of the manuscript, your rebuttal letters, and the editorial decision letters. **When submitting the final version of your manuscript please indicate whether you opt in to transparent peer review.** In the interest of confidentiality, we allow redactions to the rebuttal letters and to the reviewer comments. If you are concerned about the release of confidential data, please indicate in the cover letter what specific information you would like to have removed; we cannot incorporate redactions for any other reasons. More information on transparent peer review is available.

When you are ready to submit the final version of your manuscript, please upload the files specified in the instructions file.

Best regards,

Pep

Pep Pàmies
Chief Editor, Nature Biomedical Engineering

Nature Biomedical Engineering is a Transformative Journal. Authors may publish their research with us through the traditional subscription-access route, or make their paper immediately open access through payment of an article-processing charge. More information about publication options is available.

You may need to take specific actions to comply with funder and institutional open-access mandates. If the work described in the accepted manuscript is supported by a funder that requires immediate open access (as outlined, for example, by Plan S) and your manuscript was originally submitted on or after January 1st 2021, then you will need to select the gold OA route. Authors selecting subscription publication will need to accept our standard licensing terms (including our self-archiving policies), and these will supersede any other terms that the author or any third party may assert apply to any version of the manuscript.

Rebuttal 1

Sept 30, 2022

Dear Chief Editor, Nature Biomedical Engineering, Dr. Pep Pàmies, and the 3 Reviewers,

First of all, on behalf of the co-authors, I would sincerely like to thank you all for the comprehensive review of our manuscript and for your excellent suggestions. We have addressed all of your comments to the best of our ability and believe that the additional data and revised text make substantial contributions to improving the quality and strength of our manuscript.

We have generated quite a lot of additional data during the review process and have made changes to the text throughout the manuscript. We have removed highlighting of the new text because it became too distracting for the reader.

Hence, below I summarize the figures with their corresponding data changes. There are corresponding text changes in the introduction, results section, discussion and materials and methods.

Figure 1: unchanged

Figure 2:

- panel b added

We included virus titer calculations as requested.

Figure 3:

- panel a is modified (explanation of NovB2 use)
- panel b added (titer calculation for virus using sense and antisense vectors, with and without NovB2 has been included as requested by the reviewers)
- panel d modified (explanation of Tax and NovB2 use)
- panel e added (titer calculation for virus for sense and antisense lentiviral vectors, with and without Tax, and in combination with NovB2 has been included as requested by reviewers)

Figure 4: (complete figure was changed)

This figure presents in vitro tests performed with T cells generated with sense versus antisense vectors. We tested both anti-PSMA and anti-CD19 CARs with either luciferase or mCherry under NFAT as inducible gene-cargo. The rationale for these experiments was to better prove the presence of transcriptional interference in the sense orientation vector which causes an important decrease in the levels of inducible gene-cargo that can be produced as compared to for the antisense configuration.

Figure 5: (complete figure was changed)

In Fig. 5 we present in vivo data that supports our in vitro findings in Fig. 4 (i.e., higher levels of inducible gene cargo produced for T cells transduced with the dual antisense vector). In addition, we show that the presence of NovB2 and Tax in the culture media during lentivirus production does not impact the function of engineered T cells.

Figure 6: (this was the previous Fig.5 and it was integrated by additional in vitro tests for HPK1 knockdown T cells)

- panel c added (titer calculation for virus with or without NovB2 and TNF α)
- panel i added (cytotoxicity assay for HPK1 knockdown T cells).
- panel j added (IFN γ production assay for HPK1 knockdown T cells).
- Panel k added (proliferation assay for HPK1 knockdown T cells).

Figure 7: (this was the previous Fig.6 and it was integrated with titer calculations and replicates for the panel d experiment)

- Panel a, right (titer calculations included)
- Panel b, left (titer calculations included)
- Panel d, right (addition of three independent replicates demonstrating knockdown of the TCR-alpha chain; we do not show abrogation of TCR mispairing as this is just a proof-of-principle for use of our dual antisense vector and lentivirus production protocol having a potential clinical application)

Supp. Fig.1: unchanged

Supp. Fig.2: entire figure added

We report five independent experiments for sense versus antisense promoter interference experiments in support of Fig. 1

Supp. Fig.3:

- Panel a, top right (virus titer calculation added)
- Panel b, top right (virus titer calculation added)

Supp. Fig.4: entire figure added

We cloned dual sense and anti-sense vectors encoding a constitutively expressed TCR and gene-cargo under 6xNFAT including IL-2 and mCherry. We tested differences in effector function and gene-cargo expression levels for T cells engineered with dual sense versus antisense vectors. The findings (better gene-cargo expression in the context of the anti-sense design) support data in Figs. 4 & 5.

Supp. Fig.5: panels a-f are from previous Fig 4, panels g-j are new data

The data presented in a-f provide in vivo proof of principle for dual anti-sense lentiviral vector engineered T cells co-expressing luciferase under NFAT.

The data presented in g-j include in vivo tumor control for dual antisense vector engineered T cells expressing 4G anti-CD19 CAR T cells. These data support findings in Fig. 5. We further show that the addition of Tax and NovB2 during virus production has no impact on T cell activity.

Supp. Fig.6: entire figure added

In this figure we show that the addition of Tax and NovB2 during viral production does not have an impact on T cell activity both in vitro and in vivo (as reported in Fig.5 and Supp.Fig.5)

Supp. Fig.7: figure comprises data from previous Fig.5, panel c is new

With data in this Supp. Fig. we support the use of TNF α to boost lentiviral titers (as a substitute for Tax).

Panel c shows viral titers for sense vectors produced with or without TNF α .

Reviewer #1 (Report for the authors (Required)):

Reichenbach and colleagues designed an inverted lentiviral transfer vector, which combined with renewed strategies to enhance virus production significantly improves the efficiency of T cell engineering to provide these cells with gene cargo. These findings represent highly relevant and timely progress to the field of adoptive T cell therapy enabling the making of more effective cellular products to treat solid tumors.

Many thanks for your thorough and thoughtful review of our manuscript! We have heeded all of your comments, generating many additional new data and modifying our text accordingly, and we believe that this has significantly improved the quality of our manuscript.

Recommendations:

(1) The introduction is biased towards CAR T cells. Given the relevance of co-engineering for both CAR and TCR T cells alike, as well as data presented in Figure 5, I recommend to include TCR T cells from the start along with relevant references regarding co-engineering of the latter cells.

Thank you for this comment. You are right, we have not sufficiently addressed TCR-T cells in the context of coengineering and their very important role in cancer immunotherapy. We have now done so throughout the manuscript, including in the introduction and in our results section. Indeed, we have generated additional data for T cells engineered with a clinically relevant HLA-A2 restricted NY-ESO-1₁₅₇₋₁₆₅ specific TCR and various inducibly expressed gene-cargo from our antisense vector including IL-2, a miR-based shRNA knockdown, mCherry and luciferase. Please refer to Fig. 6 and Supp. Fig. 4. We demonstrate significantly higher levels of inducible gene-cargo expression by TCR-T cells engineered with an antisense versus sense lentivirus vector.

(2) Conclusions regarding anti-sense being the preferred design for transfer vectors are fair, yet authors should acknowledge that the head-to-head comparison of the 3 designs is challenged by the non-ability to use PA sites in the sense design. In addition, the use of different PA sites (BGH vs synthetic) and use or not of a transcription pausing site further complicates direct comparisons. Authors should discuss this.

We thank the reviewer for this comment. We have now directly addressed in the discussion the fact that we cannot do a complete 'head-to-head' comparison of the 3 designs. Indeed, it is not possible to use multiple PA sites in the sense design because this will abrogate virus production,

and it is not possible to replace the 3'LTR which acts as a PA site because it also contains elements needed for viral integration.

(3) With respect to the NOVB2 and TAX-based strategies to overcome low LV titers when using anti-sense vectors, it would provide surplus value when authors present measurements of actual titers rather than using eGFP as a surrogate marker.

We thank the reviewer for this comment. We have now included titer values, expressed as Transducing Unit (TU)/mL, throughout the manuscript.

Specifically, for TAX, and again making direct comparisons difficult, authors should justify why they switch to a CMV promotor and leave out Gene B in their tests.

Thank you for this comment. We have now included additional data demonstrating the use of Tax in the context of dual sense and antisense vectors. Please refer to Figs. 2 & 3. The single gene vector experiments have been moved to Supp. Figs 2 & 3. (The rationale for initially only using a single gene cassette for the experiments is because the NFAT promoter is inactive during virus production and thus does not contribute to dsRNA formation. However, it makes more sense to be consistent in comparing the dual gene vectors.)

(4) When following up the NOVB2 and TAX strategies with in vitro and in vivo testing of CAR T cells, authors need to include a control using single CAR vectors. Reason being: when using 'gene additives' in the HEK production phase, one formally cannot exclude that such additives are passed onto T cells potentially affecting expression and function of the CAR. In addition, the inclusion of CD19 CAR T cells vs CD19-positive targets (and using the PSMA CAR as a negative control) would provide additional robustness to findings.

We thank the reviewer for these thoughtful comments and suggestions which we have addressed experimentally.

We have now performed both in vitro and in vivo experiments comparing both TCR- and CAR- T cells engineered with antisense vectors in which the lentivirus was produced +/- NovB2 & Tax. We observed no differences in activity levels of the transduced T cells and conclude that there is no pseudo-infection taking place, or at least none that alters the activity levels of the engineered T cells. Please refer to Fig. 5 and Supp. Figs. 5 & 6.

To address your second comment regarding the use of a CD19 model, we have performed two additional in vivo studies, one with mice inoculated with CD19⁺Bjab tumors and the other with mice inoculated with PC3-CD19⁺ tumors (the PC3 cell line was gene-modified to express CD19), and ACT with anti-CD19 CAR T cells as well as anti-PSMA CAR-T cells as a negative control. Please refer to Supp. Figs. 5 & 6. We show specific tumor control by the anti-CD19 CAR-T cells. Moreover, in the context of both these tumor models we compare T cells engineered with virus produced +/- NovB2 & Tax as mentioned above (and show no differences in tumor control). Additionally, we demonstrate specific gene-cargo coexpression (luciferase under 6xNFAT) in the TME for the tumor-specific anti-CD19 CAR-T cells but not the anti-PSMA CAR-T cells in the context of the Bjab tumor model.

(5) The inclusion and testing of siRNA reads as a 'turning point'. I would rather first see an optimal protocol for transgenic cargos. Authors should consider two main parts, first finalizing the protocol for transgenes, second go into knock-down applications. Along these lines, it would be better to substantiate part 1 with 'real' gene cargo (i.e., IL12).

We thank the reviewer for this nice suggestion. We have tried to better distinguish between the results for the gene-cargo versus shRNA knockdown work. We have also performed additional experiments for TCR-engineered T cells with both IL-2 and mCherry as gene-cargo (Supp.Fig.4).

Leave out knock-down of TRAC as this is now combined w/ eGFP rather than a TCR (consequently not assessing the effect on TCR mis-pairing) and focus on just 1 knock-down example, being Hpk-1. Regarding the 2nd part (as has been done for the first part), include in vitro function of T cells, not trivial in the context of Hpk1 deficiency and its effect on TCR signaling.

We understand and respect this opinion. However, we have retained the TCR-alpha chain knockout not to explore the concept of TCR chain mispairing, but rather as an additional proof-of-principle for the ability of our strategy (i.e., dual inverted vector and use of TNF α + NovB2 during lentivirus production) to address 'difficult to produce' lentivirus such as ones including stem-structures. We hope that you agree and that our edits make the reasoning/purpose of the experiment more apparent.

With regards to HPK1, we have performed additional in vitro tests for TCR-T cells +/- HPK1 knockdown including target cell killing, IFN γ production and proliferation assays (Fig.6 i,j&k). Notably, aside from higher proliferation of HPK1 knockdown CD8⁺ TCR-T cells in target tumor cell coculture assays, we did not observe improved effector function by HPK1 knockdown TCR-T cells. This is in contrast to previous literature showing improved T cell function upon pharmacological inhibition or knockout of HPK1 (cited in the manuscript). We have not explored this further because it goes beyond the scope of our manuscript which is really to show that that our vector 'works' for knocking down target genes by miR-based shRNA.

Minor points:

- Manuscript is well written, although here and there at an educational level. Please make it less tutorial (many explanations between brackets are no necessity).

We thank the reviewer for this comment. We have removed many of the explanations in brackets. Please note that we really set out with the goal of allowing a reader with limited knowledge of vector design and gene engineering to understand our work. We hope that this is the case, but in a less tutorial manner.

- Discussion: avoid repetition of introduction. Rather discuss limitations of proposed methods, and what experiments would be needed to implement these methods in the clinical setting.

We have revised parts of the discussion and hope that they meet your satisfaction.

Also, authors should discuss why CD4 T cells appear more prone for these transfer vector-mediated transductions when compared to CD8 T cells?

Interestingly, we almost always observe higher transduction efficiency of primary human CD4⁺ T cells than CD8⁺ T cells, even with simple lentiviral sense vectors encoding only a CAR, for example. Based on your comment we thought that we would dig into the reason for this. In our experiments we have used lentivirus comprising the VSVG envelope protein and hence measured the level of LDL-R on the surface of both CD4⁺ and CD8⁺ T cells following 24-hour activation by anti-CD3 and anti-CD28 antibody coated beads, plus hIL2 50U/mL. As shown in **Fig. 1** below we did not observe significant differences, both in terms of percentage and MFI of LDL-R.

Figure 1: Analysis of FACS LDL-R expression values (percentage on the left and MFI on the right) on both CD4⁺ and CD8⁺ T cells after 24h activation protocol (right before transduction with lentiviral supernatant) (n=3 healthy donors derived T cells).

Another possible explanation is the phenotype of the T cells at the time of transduction as we do not enrich for naïve T cells for our experiments. We evaluated the proportion of naïve (N), central memory (CM), effector memory (EM) and Terminally-differentiated T cells (T-EMRA) isolated from freshly processed PBMC (**Fig. 2**). Indeed, we observe differences between CD8⁺ and CD4⁺ T cells but further studies are needed to understand if this impacts transduction efficiencies. We have not included these data in the manuscript.

Figure 2: Flow cytometric-based phenotypic analysis of CD4⁺ and CD8⁺ T cells purified from the PBMCs of five healthy donors and activated for 24 hours. (CM=CCR7⁺CD45RA⁻; EM=CCR7⁻CD45RA⁺; Naive=CCR7⁺CD45RA⁺; T-EMRA=CCR7⁻CD45RA⁺).

We hope to eventually find a satisfying explanation as to why it is 'easier' to lentivirally transduce CD4⁺ than CD8⁺ T cells.

- In line with the above recommendation to split Manuscript into 2 parts, Figure 5 could go supplementary.

As mentioned above, we tried to better divide the manuscript into a "transgene section" and "miR-shRNA section". The order of Main and Supp. Figs. has been modified according to the manuscript flow.

- When measuring the 2nd gene product upon activation w/ CAR T cells (Figure 4c), it would be better to replace PMA/I by an anti-CAR stimulus.

We have made important changes to Fig.4 and now include activation of the engineered T cells with target cell lines. PMA-Ionomycin stimulation has only been used as a control to demonstrate maximum gene-cargo expression levels.

- Figures: F4: include the NovB2 and TAX methodology in cartoons; F4c: CAR transductions are normalized to 40% (in text it states 50%); F6a has little additional value over F5b; F6c, cartoon, CAR should be replaced by eGFP; and SF1, axes w/ dot plots are not annotated.

We thank the reviewer for the comments and we changed the figures, correcting the annotated mistakes.

Reviewer #2 (Report for the authors (Required)):

This is an interesting manuscript in which the authors explore the optimal configuration of lentiviral vectors to express two transgenes (constitutive x2 or constitutive and inducible post activation), or a siRNA and a transgene. The authors use a very methodological approach and the results are convincing although the 'n' of most experiments seems very low. However, my biggest concern centers around novelty and innovation.

We sincerely thank the reviewer for their thorough review of our manuscript and helpful comments. We hope that with the additional data generated during the review process, along with improved explanations in our manuscript, that we can convince you of the importance of our work to the field and its novel aspects.

Major concerns

1) No new promoters or conceptually new vector designs are being explored or being developed. For example, one of the developed lentiviral vectors enables the expression of siRNAs and a

transgenes. Retroviral vectors with the same capabilities were developed more than a decade ago (PMID: 23250361, PMID: 21673345).

It is true that vectors have previously been described which allow co-expression of shRNA with additional transgenes. It is important to note, however, that the shRNAs are usually expressed from a Pol III promoter like U6 and this approach is well-known to be toxic to transduced cells due to an 'overcharging' of the RNA interference machinery. Thus, the development of a strategy (i.e. vector design and lentivirus production protocol) allowing the use of Pol II promoters to independently drive gene knock-downs, either constitutively or inducibly, and in the absence of transcriptional interference, is an important accomplishment.

Indeed, in the two cited publications, expression problems will be encountered for the following reasons:

- a) The presence of 4 shRNA structures will impair viral production due to RNA processing by DICER thus resulting in low titers.
- b) Post transgene integration in transduced cells, the TCR α -chain messenger RNA will be destabilized by the presence of the co-transcribed shRNAs lowering the final expression level.
- c) Post transgene integration in transduced cells, LTR-driven TCR α -chain will be transcribed over the hPGK-TCR β -chain to reach the 3'LTR for poly-adenylation creating transcription interference as described in our results section for **Fig.1**.

A scheme is provided below to summarize the described effects.

HEK-293T

T cells

T cells

We hope you agree that, taken together, our approach is an important advance for lentiviral vector mediated, independent, multi-gene cellular engineering. In summary, we have described the development of a dual antisense vector allowing *independent* expression of 2 genes in the absence of transcriptional interference. However, low-titers for this lentiviral vector design forced us to find creative solutions, including the use of NovB2, Tax and TNF α during lentivirus production. We believe that our strategy will be useful for many labs as well as for GMP application, in particular the use of TNF α in the culture media is a very appealing strategy.

2) It seems that many experiments were only performed once or only in technical replicates (e.g., Figs 1-

We thank the reviewer for this comment. We have revised the figure legends to specifically state the number of *independent experiments* that were performed, as well as the number of technical replicates. In our graphs, each symbol represents the mean of 2-3 technical replicates of an independent experiment. For in vitro assays, the minimal number of independent experiments is 3. Notably, for each experiment we use T cells from independent anonymous healthy donors (we always work with fresh buffy coats and freshly transduced T cells that have never been frozen).

3). It is unclear how many different human donors were used to conduct the experiments.

This has now been specified in all figure legends.

Minor concerns

1) While titration experiments with lentiviral vectors are performed, no formal titers are calculated. This is critical to assess if the developed lentiviral constructs are suitable for clinical translation.

We thank the reviewer to highlight this point and we added the titer calculations, expressed as Transducing Unit (TU)/mL, for each virus produced with/without NovB2, Tax and TNF α in the following figures: Figure 2 panel b, Figure 3 panel b and e, Figure 6 panel c, Figure 7 panel a (right), panel b (right), Supplementary Figure 2 panel a (right) and panel b (right), Supplementary Figure 7 panel c.

Reviewer #3 (Report for the authors (Required)):

In their manuscript, Reichenbach and colleagues describe the design of an inverted lentiviral vector that contains one promoter to constitutively express one gene (e.g. a chimeric antigen receptor (CAR) to direct T cell activity against tumor cells expressing target antigens) and a second promoter that acts in an inducible manner by responding to signals that derive from activation of the constitutively expressed CAR construct. This work is reminiscent of earlier descriptions of armored CARs and TRUCK concepts that also used NFAT response elements coupled with the minimal IL2 promoter (as the authors also correctly reference).

The authors show some of the optimization steps they used to decrease non-specific activation of the inducible gene cassette component of this newly designed vector system. In the abstract, the authors promise reduced lentiviral production costs as this is a single vector system that can be used to generate higher titers. As also referenced by the authors (e.g. refs 34 and 35), other single vector systems for such purposes were published earlier. To address the claim that higher lentiviral vector titers can be achieved with the vector system described in this manuscript. The authors do show that co-delivery of NovB2 and Tax (later substituted for TNF α) proteins can be used to increase transduction efficiencies of the lentiviral vectors described here, however, this effect is difficult to assess in the manner that the data is presented. In several instances, the authors provide volumes of lentiviral vector supernatants that were applied for the transduction experiments.

The authors should provide some calculation of vector titers for comparison to other established systems. This is especially important as the quality / infectivity of different lentiviral vector productions can vary due to several factors.

In general, such approaches to develop synthetic vector systems for improved modification of immune cells is of high interest and could have impact on the fields of immunotherapy, virology and retroviral vector development and could even become clinically relevant if shown to be robust. Indeed, the authors present data showing transduction of primary human T cells with the inverted lentiviral vector that was designed to express CARs and that the CAR T cells were active against

cancer cells in vitro and in vivo. While these findings might not be surprising as CAR constructs known to function were used, it does provide proof-of-principle that the current vector system can be successfully used to generate functional CAR T cells. In addition to the points mentioned already, please find some additional comments that should be addressed:

We sincerely thank the reviewer for their thorough review of our manuscript and helpful comments. Based on your suggestions, the text has been improved to hopefully provide clearer descriptions/explanations, titer calculations have been included, and we have performed many additional experiments, including further in vivo testing of T cells engineered with the antisense vector.

Major points to be addressed:

1. In figure 1, representative flow cytometric analyses are shown. It would be more informative and appropriate to include graphs that show the results for all experiments accomplished (i.e. more than the one experiment shown in panels a, b and c). This would provide the reader a better chance to evaluate the presented data. Also, in some figure legends, the authors write “representative” without providing further information – these data are representative of how many experiments?

We thank you for this suggestion. The Fig.1 dot plots are representative of five independent experiments. We have now included all of the dot plots from the additional experiments in Supp.Fig.2.

In all figure legends we have now stated the number of *independent experiments* (i.e. not technical replicates) that were performed.

2. As mentioned above, the authors describe transduction with volumes (μ l) of lentiviral vector supernatants. These should rather be shown as multiplicities of infection to apply similar amounts of infectious particles and/or quantification of lentiviral capsid or RNA. Otherwise, it is difficult to judge and compare the effectiveness of the lentiviral vector preparations.

We thank the reviewer for this comment. We have now included titer calculations, expressed as Transducing Unit (TU)/mL, for lentivirus produced with/without NovB2, Tax and TNF α in the following figures: Fig.2b, Fig.3b&e, Fig. 6 c, Fig.7a&b, Supp. Fig.2a & b, Supp. Fig. 7 c.

3. What is the authors' rationale for using the 30 μ l lentiviral vector supernatant amounts for comparison to claim improved transduction of Jurkat cells with the anti-sense vector in combination with NovB2 and Tax? This seems arbitrary as little or no improvement to transduction levels are seen with the combination when higher volumes of the lentiviral vector supernatants are used. The authors could add a sentence in the appropriate section of the results to make this more clear.

To assess the virus titer, increasing amounts of viral supernatant are used to transduce a fixed amount of target cells (eg.100.000 cells/ condition; in our paper the target cells used are Jurkats).

The reporter transgene (eg. eGFP, TCR, CAR...) is then quantified in all the assayed dilution conditions. By using the relative percentage of positive cells for the transduced reporter gene, the number of transduced cells will correspond to the amount of active transducing viral units in the supernatant. However, this assumption is valid only if one copy of transgene per cell is achieved. For this reason, the calculation of the transducing unit/ml is done using one point of the viral supernatant dilution curve at which the level of transduction is around 20-30%. The rationale of using the 30 μ L dilution to calculate the titer in our experiments was because at that point our transduced cells cannot have more than one copy of transgene/target, thus making the titer calculation valid. We have updated our methods section to include viral titer calculations.

4. In figure 4, a comparison with the sense lentiviral vector with regard to lentiviral vector supernatant titers of the CAR vectors seems to be missing.

For this experiment our goal was to normalize CAR/TCR expression levels in order to perform functional tests to explore transcriptional interference of the inducible gene cargo. We have now added additional data generated during our revision, including a functional comparison of sense versus antisense vectors both in vitro (Fig.4, Supp.Fig.4) and in vivo (Fig.5, Supp.Fig.5).

In figure 4c, how do the authors explain the strong mCherry background expression in the unstimulated CAR T cells?

Additional data has been added to Fig. 4 and Fig. 4c has been replaced with a different experiment. We believe, however, that the background is due to the scFv used (FMC63) to target CD19 which has previously been described to drive tonic signaling (Nat Med. 2015 Jun; 21(6): 581–590), which can account for the mCherry background in unstimulated CAR T cells.

5. In the description of the data shown in figure 5, the authors write “important increase in titer”, but do not actually show any titers for the lentiviral vectors. These should be calculated and shown.

Thanks for pointing this out. We have now added titer calculations throughout the manuscript.

Minor points to be addressed:

1. In many instances throughout the manuscript, the authors write “lentivirus mediated”. I think this would be more correct as “lentivirus vector mediated”, as a vector derived from the lentivirus is used.

Thank you for this good suggestion. We have modified the text to use this term.

2. Sometimes the authors write “ml” in the figure legends describing the amount of lentiviral vector supernatant used. This is presumable μ l.

Thanks, the symbol font was lost somehow...The “mL” was corrected to “ μ L”.

3. In supplemental figure 1, the labels for the flow cytometry plot axes are missing (i.e. eGFP and mCherry).

We have corrected Supp. Fig.1. Thanks for catching this mistake.

4. The manuscript reads mostly ok, but does need some editing for punctuation and language use (e.g. “ameliorate” is often used incorrectly).

Thank you. We have carefully edited the manuscript. And you are right about the incorrect use of the word ameliorate – it has been replaced with improved or enhanced, etc.

Rebuttal 2

Response to reviewers:

Reviewer #1 (Report for the authors (Required)):

Reichenbach and colleagues have profoundly and accurately addressed and/or rebutted all of my concerns that were raised in response to their original contribution. The revised manuscript shows improved quality and quantity of data, as well as a better overall structure, flow of reading and discussion of findings.

We are pleased to have fully rebutted all the concerns of Reviewer #1 and we sincerely thank them again for their very careful review of our manuscript and excellent comments and suggestions.

Reviewer #2 (Report for the authors (Required)):

I would like to thank the authors for submitting a substantially revised manuscript that addresses all my concerns. I would like to congratulate the authors to their important study.

We are happy to have addressed all concerns of Reviewer #2 and we sincerely thank them again for their very thoughtful review of our manuscript and excellent feedback and suggestions. We also thank Reviewer #2 for their kind comment on our final manuscript!!

Reviewer #3 (Report for the authors (Required)):

In the revised manuscript, Reichenbach and colleagues have addressed most of the concerns raised during the first round of review by adding additional data and further editing the text of the manuscript. In short, the manuscript is greatly improved, but there remain some issues to address.

We sincerely thank Reviewer #3 again for their careful review of our manuscript and for their excellent comments and suggestions. As per our responses below, we believe that we have now addressed any remaining issues.

Major technical point to address: The authors addressed the question about the strong mCherry background expression in unstimulated T cells shown in the original figure 4c by replacing this data with other experiments, but I think this might have been important data as it is always best to know potential drawbacks of an approach or further areas of possible improvement (such as

to overcome leakiness of an inducible promoter). In the revised figure 4 panels g and h, non-stimulated controls to quantify the background mCherry reporter expression from the CAR-T cells seems to be missing.

We thank the reviewer for highlighting this point. We have re-introduced the previous Fig.4c (as Supplementary.Fig.5b) and address the issue of background expression levels of genes under 6xNFAT in non-activated CAR-T cells.

In the revised Figure 4 we have now included control data for unstimulated T cells (Fig.4g) and address the issue of background expression of gene-cargo under 6xNFAT in non-activated CAR T cells.

We have highlighted all modified text.

Minor (but still important points to address): Although the authors now use “lentivirus vector” in some instances in the modified manuscript, they do still quite often use the term “lentivirus” throughout the manuscript, which, is incorrect as the system described in this work is a lentivirus vector system. Thus, this still needs to be corrected in many instances.

We thank the reviewer for this comment. We have now replaced lentivirus by lentivirus vector throughout the text (we have highlighted these changes).